## REPORT

# Wapl releases Scc1-cohesin and regulates chromosome structure and segregation in mouse oocytes

Mariana C.C. Silva[1] , Sean Powell[2]* , Sabrina Ladstätter[2]* , Johanna Gassler[2]* , Roman Stocsits[1] , Antonio Tedeschi[1], Jan-Michael Peters[1] , and Kikuë Tachibana[2,3]

**Cohesin is essential for genome folding and inheritance. In somatic cells, these functions are both mediated by Scc1-cohesin, which in mitosis is released from chromosomes by Wapl and separase. In mammalian oocytes, cohesion is mediated by Rec8-cohesin. Scc1 is expressed but neither required nor sufficient for cohesion, and its function remains unknown. Likewise, it is unknown whether Wapl regulates one or both cohesin complexes and chromosome segregation in mature oocytes. Here, we show that Wapl is required for accurate meiosis I chromosome segregation, predominantly releases Scc1-cohesin from chromosomes, and promotes production of euploid eggs. Using single-nucleus Hi-C, we found that Scc1 is essential for chromosome organization in oocytes. Increasing Scc1 residence time on chromosomes by Wapl depletion leads to vermicelli formation and intra-loop structures but, unlike in somatic cells, does not increase loop size. We conclude that distinct cohesin complexes generate loops and cohesion in oocytes and propose that the same principle applies to all cell types and species.**

## Introduction

Meiosis is a specialized cell division in which DNA replication is followed by two rounds of chromosome segregation, producing haploid gametes. Reciprocal recombination of maternal and paternal homologous chromosomes (homologues) produces physical linkages that manifest as chiasmata on bivalent chromosomes in meiosis I. Maternal and paternal centromeres of homologues segregate in meiosis I and sister centromeres disjoin in meiosis II. In mammals, oocyte formation is initiated during fetal development, with meiotic DNA replication and recombination occurring before birth, but is only completed from puberty onwards, when oocytes undergo the meiosis I division at ovulation (Hassold and Hunt, 2001).

Homologous chromosomes assemble into bivalents, which are held together by cohesin complexes. These are thought to mediate cohesion by entrapping sister DNAs (Haering et al., 2008) and are essential for meiotic chromosome segregation. Cohesin complexes are formed by a heterodimer of Smc3 and either Smc1α or Smc1β, which is bridged by an α-kleisin that can be Rec8, Scc1, or Rad21L in mammalian germ cells (Rankin, 2015; Revenkova and Jessberger, 2006). Rec8-cohesin is essential for chromosome arm and centromere cohesion, while Scc1-cohesin is dispensable for cohesion in meiosis (Tachibana-Konwalski et al., 2010). In contrast, Scc1 is the only α-kleisin (Lee et al., 2002) in mammalian somatic cells, where it mediates both cohesion and long-range chromosomal cis interactions that can be detected by Hi-C as loops and topologically associating domains (TADs; Gassler et al., 2017; Haarhuis et al., 2017; Schwarzer et al., 2017; Wutz et al., 2017; Rao et al., 2017). Whether Scc1-cohesin also has a function in oocytes or if it is maternally deposited to establish cohesion after fertilization in zygotes is unknown (Ladstätter and Tachibana-Konwalski, 2016).

Cohesin can actively be released from DNA by Wapl or the protease separase (Nasmyth et al., 2000; Peters and Nishiyama, 2012). Separase-mediated cleavage of Rec8 releases chromosome arm and centromeric cohesion to trigger homologue disjunction in anaphase I and sister centromere disjunction in anaphase II, respectively (Kudo et al., 2006; Tachibana-Konwalski et al., 2010). In somatic cells, Wapl releases cohesin from chromosome arms in mitotic prophase, and to a lesser extent throughout interphase (Gandhi et al., 2006; Kueng et al., 2006; Tedeschi et al., 2013; Haarhuis et al., 2013). In budding yeast, *Caenorhabditis elegans* and *Arabidopsis thaliana*, Wapl has roles in releasing cohesin from meiotic chromosomes and is required for proper meiosis (Challa et al., 2016, 2019; Crawley et al., 2016; De

..................................................................................................................................

[1]Research Institute of Molecular Pathology, Vienna BioCenter, Vienna, Austria; [2]Institute of Molecular Biotechnology of the Austrian Academy of Sciences, Vienna BioCenter, Vienna, Austria; [3]Department of Totipotency, Max Planck Institute of Biochemistry, Martinsried, Germany.

*S. Powell, S. Ladstätter, and J. Gassler contributed equally to this paper; Correspondence to Kikuë Tachibana: kikue.tachibana@imba.oeaw.ac.at; Jan-Michael Peters: peters@imp.ac.at; M.C.C. Silva's present address is Instituto Gulbenkian de Ciência, Oeiras, Portugal; A. Tedeschi's present address is The Francis Crick Institute, London, UK.

et al., 2014). In budding yeast and *A. thaliana*, Wapl releases Rec8-cohesin during prophase I (Challa et al., 2019; De et al., 2014). In contrast, in *C. elegans* Wapl only releases cohesin complexes containing the α-kleisin subunits COH3/4 and does not regulate Rec8-cohesin during meiotic recombination (Crawley et al., 2016). Rec8-Stag3-cohesin, ectopically expressed in human somatic cells, is susceptible to Wapl-dependent release and protection by the Wapl antagonist sororin, suggesting that this complex can also be a target of Wapl (Wolf et al., 2018). However, whether Wapl is required for mammalian meiosis and whether it contributes to release of chromosomal Rec8, Scc1, or both in oocytes is not known.

## Results and discussion

### Wapl is required for proper chromosome segregation of meiosis I oocytes

To address Wapl's role during meiosis, we used a conditional genetic knockout approach based on *(Tg)Zp3*-Cre to delete floxed alleles of *Wapl* (also known as *Wapal*) in growing phase oocytes (Fig. 1 A; Lewandoski et al., 1997; Tedeschi et al., 2013). In this mouse model, *Wapl* is unperturbed during meiotic DNA replication and recombination in fetal oocytes and deleted in the 3 wk before oocyte maturation. Crossing *Wapl*$^{fl/fl}$ *(Tg)Zp3*-Cre females to wild-type males resulted in *Wapl*$^{Δ/+}$ offspring, demonstrating efficient deletion of floxed alleles. Whether *Wapl*$^{fl/fl}$ *(Tg)Zp3*-Cre females are fully fertile is not clear because larger numbers of crosses would have to be analyzed to assess this, but litter production suggests that the meiotic divisions can proceed without Wapl.

To analyze the effects of Wapl loss on meiosis I, we isolated *Wapl*$^{fl/fl}$ and *Wapl*$^{Δ/Δ}$ oocytes from *Wapl*$^{fl/fl}$ and *Wapl*$^{fl/fl}$ *(Tg)Zp3*-Cre females, respectively. Control *Wapl*$^{fl/fl}$ oocytes progressed through the meiosis I division and extruded polar bodies in 8 h 20 min ± 47 min (Fig. 1 B). *Wapl*$^{Δ/Δ}$ oocytes extruded polar bodies at 7 h 50 min ± 50 min (Fig. 1 B), suggesting that the first division occurs with mildly faster kinetics (*, P = 0.0286). To examine the dynamics of chromosome segregation, we microinjected *Wapl*$^{fl/fl}$ and *Wapl*$^{Δ/Δ}$ germinal vesicle (GV)–stage oocytes with mRNA encoding H2B-mCherry to mark chromosomes and performed live-cell imaging (Fig. 1, C and D; and Videos 1, 2, and 3). Wapl depletion induced stretching of bivalents aligned at the metaphase I plate (Figs. 1 C and 3 A, and Videos 2 and 3). Lagging chromosomes in anaphase I occurred at a similar frequency for control and knockout oocytes (20 ± 13% and 24 ± 10%, respectively; P > 0.99, ns; Fig. 1 D). In contrast, chromosome bridges were not detected in *Wapl*$^{fl/fl}$ but occurred in 43% of *Wapl*$^{Δ/Δ}$ oocytes (Fig. 1 D; *, P = 0.03). These are reminiscent of anaphase bridges observed in somatic cells lacking Wapl (Haarhuis et al., 2013; Tedeschi et al., 2013). The molecular causes of these bridges are not known in any cellular system. They could be due to either topological entanglements between homologous chromosome arms or inefficient separase-mediated cleavage of excessive chromosomal cohesin, which could be resolved over time or lead to aneuploidy. Incorrectly repaired DNA breaks could also cause chromosome bridges (see below).

To determine whether Wapl protects against aneuploidy, we examined chromosome number and type in meiosis II eggs. Due

to the technical caveat that chromosome spreading can lead to chromosome loss, we considered hyperploidy (>20 dyad chromosomes) as a stringent measure of aneuploidy. The frequency of hyperploid eggs was 2.9% and 8.2% in *Wapl*$^{fl/fl}$ and *Wapl*$^{Δ/Δ}$ eggs, respectively (Fig. 1, E and F). Chromosome missegregation and egg aneuploidy increased further with age (Fig. S1, A–D), suggesting that Wapl loss exacerbates age-related defects. Precociously separated sister chromatids (PSSC) are also a measure of prospective aneuploidy because they can segregate randomly in meiosis II. PSSC was detected in 1.5% and 8.2% of *Wapl*$^{fl/fl}$ and *Wapl*$^{Δ/Δ}$ eggs, respectively (Fig. 1, E and F). This increase in PSSC is at odds with the expectation that Wapl loss prevents release of cohesin mediating cohesion and implies that Wapl depletion affected another pathway. Considering both types of chromosomal anomalies, we observed a total of 4.4% *Wapl*$^{fl/fl}$ and 16.4% *Wapl*$^{Δ/Δ}$ aneuploid eggs (Fig. 1, E and F; *, P = 0.0276). We conclude that Wapl is required for proper meiosis I chromosome segregation and promotes production of euploid eggs.

### Wapl predominantly releases Scc1 from bivalent chromosomes

To identify targets of Wapl, we examined different cohesin complexes in meiosis I by immunofluorescence microscopy. Rec8 and Smc3 localize to the inter-chromatid axis, whereas Scc1 is close to the detection threshold in control oocytes, as reported previously (Fig. 2 A, insets; Tachibana-Konwalski et al., 2010). If Wapl is releasing cohesin that mediates cohesion, then one would expect an increase in chromosomal Rec8 abundance in oocytes lacking Wapl. However, Rec8 chromosomal abundance was comparable in *Wapl*$^{fl/fl}$ and *Wapl*$^{Δ/Δ}$ oocytes (Fig. 2, A and B), suggesting that Wapl is releasing little or no Rec8-cohesin. In contrast, Smc3 chromosomal abundance increased 6.5-fold and was enriched along the chromatid axis in *Wapl*$^{Δ/Δ}$ oocytes; a similar effect was observed for Smc1α (Fig. 2, A and B). Interestingly, Scc1 also increased threefold and was enriched along chromatids in *Wapl*$^{Δ/Δ}$ oocytes (Fig. 2, A and B), suggesting that Wapl is actively releasing a cohesin complex containing Scc1-Smc3-Smc1α. Since these proteins form a 1:1:1 stoichiometric complex, it is unexpected that Wapl depletion increased the chromosomal signals of these subunits to different degrees. We suspect that this reflects different antibody affinities, and not additional effects of Wapl depletion on Rec8-cohesin, because codepletion of Wapl and Scc1 reduced chromosomal Smc3 levels to those seen in *Wapl*$^{fl/fl}$ oocytes (see Fig. 3, E and F). However, our results do not exclude the possibility that Wapl might release small amounts of Rec8, as observed in an ectopic HeLa cell expression system (Wolf et al., 2018). Expression of mRNA encoding Wapl in *Wapl*$^{Δ/Δ}$ oocytes was sufficient to decrease Smc3 abundance to levels comparable to those observed in *Wapl*$^{fl/fl}$ oocytes (Fig. S2, A and B), indicating that Wapl is directly releasing cohesin from meiosis I chromosomes.

*Wapl* deletion also altered bivalent structure (Fig. 2, A and C). In *Wapl*$^{fl/fl}$ oocytes, most bivalents have a single chiasma, and only 20% have more than one chiasma. In contrast, more than one chiasma-like structure was detectable in >60% of bivalents in *Wapl*$^{Δ/Δ}$ oocytes (Fig. 2 C). We can exclude that these additional structures are due to additional crossovers since crossover

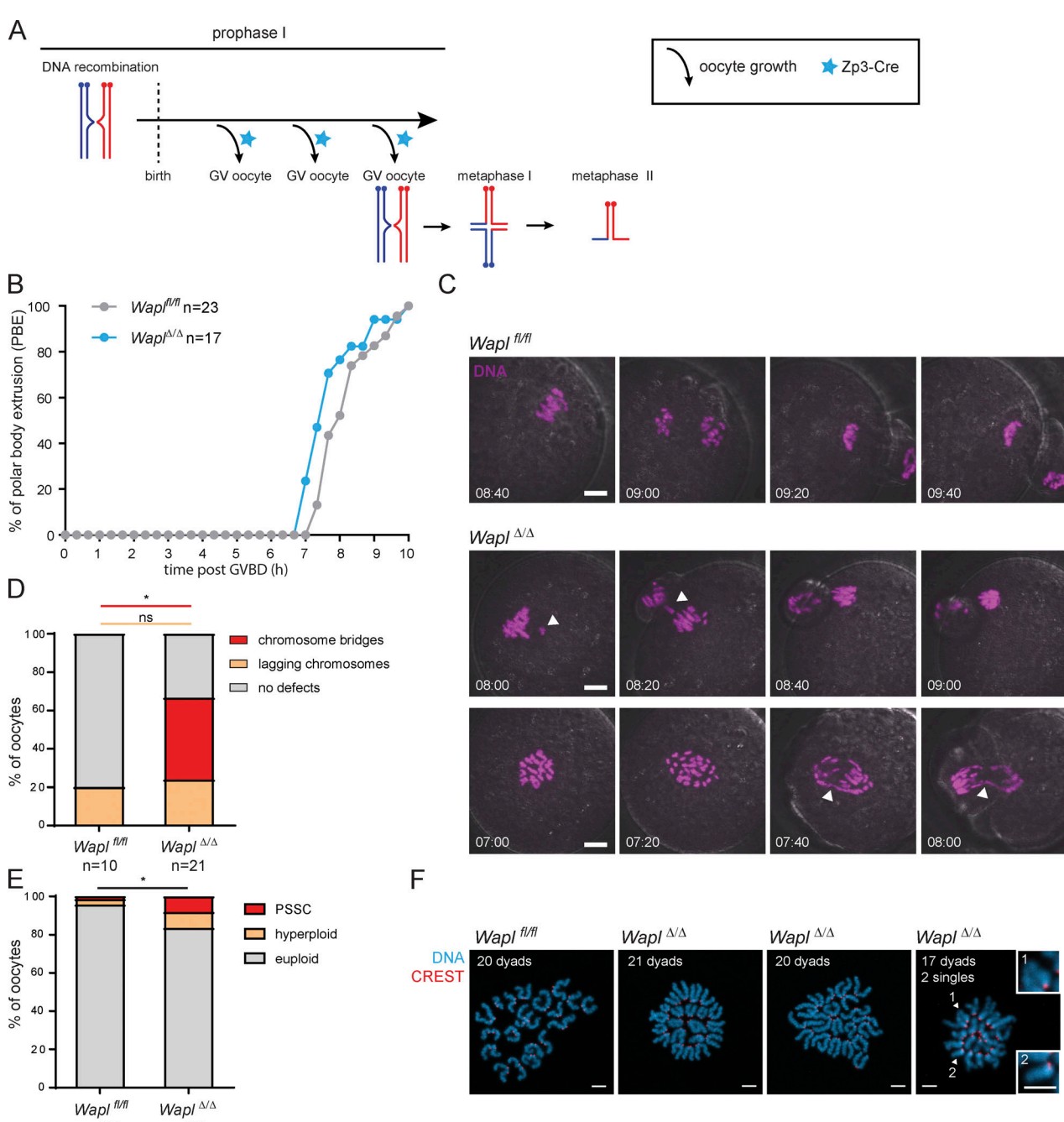

Figure 1.   **Wapl is essential for normal segregation of homologues in meiosis I. (A)** Schematic representation illustrating that activation of *(Tg) Zp3*-Cre (blue stars) leads to *Wapl* deletion after birth during the oocyte growing phases that precede meiosis I resumption. The three branching arrows represent the different cycles of oocyte growth that precede each round of meiotic divisions. The blue stars represent activation of Zp3-Cre. **(B)** The timing from GV breakdown (GVBD) to anaphase polar body extrusion (PBE) was quantified in *Wapl^fl/fl* and *Wapl^Δ/Δ* oocytes by low-resolution live-cell imaging. The number of oocytes analyzed per condition is indicated. *, P = 0.0286 (Mann-Whitney test). **(C)** Representative stills of high-resolution live-cell imaging videos showing chromosome segregation in *Wapl^fl/fl* and *Wapl^Δ/Δ* oocytes. DNA is shown in magenta. White arrowheads indicate chromosome bridges, lagging chromosomes or misaligned chromosomes. The time displayed indicates hours after GVBD. Scale is the same in all images; scale bar, 10 μm. **(D)** Quantification of chromosome segregation defects during meiosis I observed in high-resolution live-cell imaging videos of *Wapl^fl/fl* and *Wapl^Δ/Δ* oocytes. Three *Wapl^fl/fl* and three *Wapl^fl/fl (Tg) Zp3*-Cre littermate females were analyzed, and the total number of oocytes examined for each genotype is indicated in the figure. The graph shows the percentage of oocytes for each chromosome-segregation phenotype (indicated in the legend). P values were calculated using Fisher's exact test and are >0.99 for lagging chromosome defects (ns, not significant) and 0.03 for chromosome bridges (*, significant). **(E)** The number of dyads was quantified in metaphase II chromosome spreads of *Wapl^fl/fl* and *Wapl^Δ/Δ* oocytes. The metaphase II chromosome spreads were classified in euploid (20 dyads), hyperploid (>20 dyads) and PSSCs. Six *Wapl^fl/fl* and six *Wapl^fl/fl (Tg)Zp3*-Cre littermate females were analyzed, and the total number of oocytes analyzed is indicated in the figure. P value for total aneuploidy (including both presence of PSSC and hyperploidy) was calculated using Fisher's exact test and is 0.0276 (*). **(F)** Representative images of metaphase II spreads of *Wapl^fl/fl* and *Wapl^Δ/Δ* oocytes. Centromeres are shown in red and DNA in blue. The number of dyads per oocyte spread is indicated. The white arrowheads indicate single chromatids. Insets 1 and 2 show single chromatids observed in *Wapl^Δ/Δ* oocytes. Scale bar, 5 μm; inset scale bar, 5 μm.

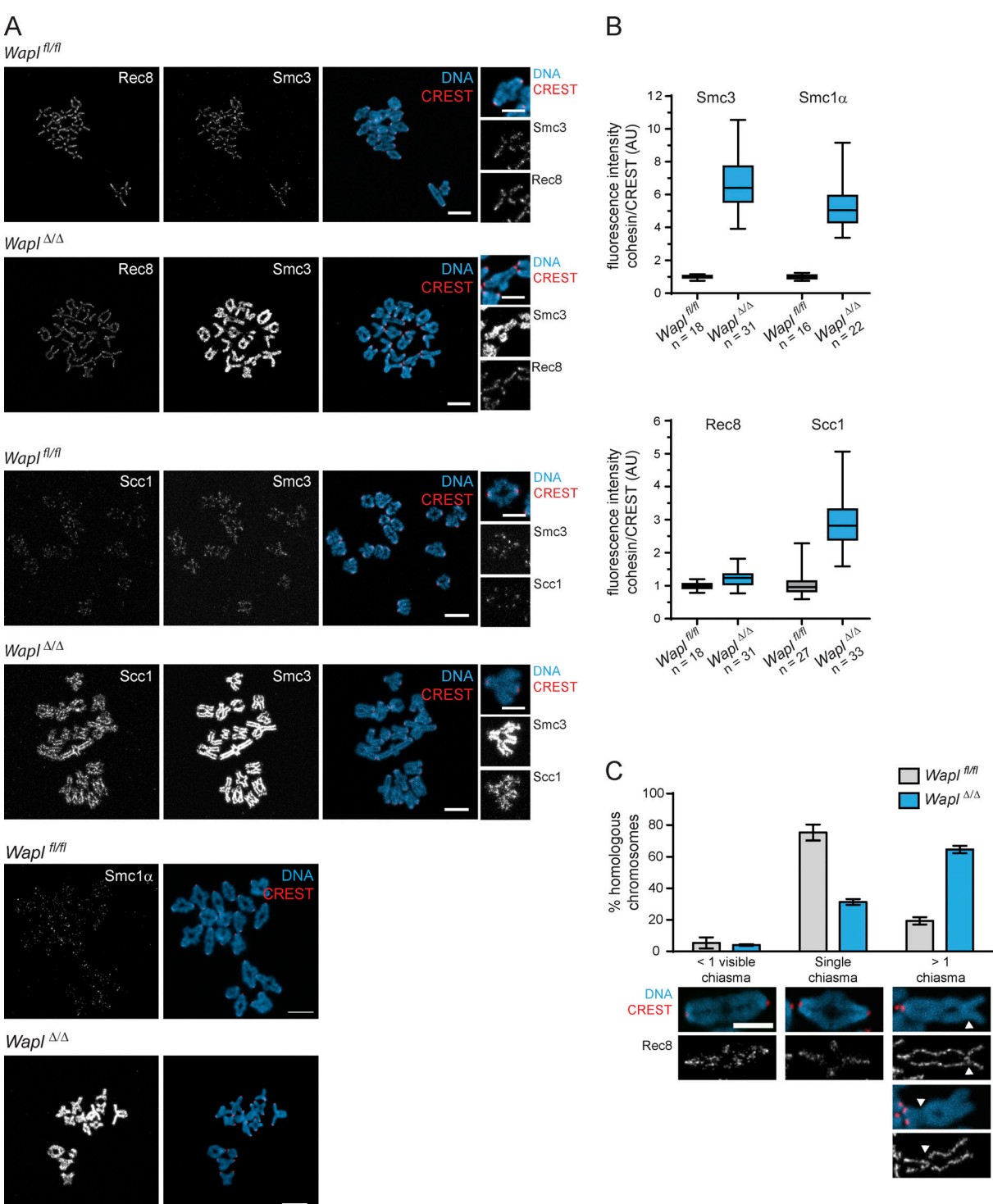

Figure 2. **Wapl predominantly controls chromatin-associated levels of Scc1-cohesin in mouse oocytes. (A)** Representative images of pro-metaphase I spreads of *Wapl^fl/fl* and *Wapl^Δ/Δ* oocytes. Centromeres are shown in red, Smc3, Scc1, and Rec8 in gray, and DNA in blue. Scale bar, 10 μm; inset scale bar, 5 μm. **(B)** Quantification of Smc3, Smc1α, Rec8, and Scc1 fluorescence intensities per bivalent in relation to CREST fluorescence intensities in *Wapl^fl/f* and *Wapl^Δ/Δ* oocytes. This ratio of fluorescent intensities is presented in arbitrary units (AU). For Smc3 and Rec8 quantifications, three *Wapl^fl/fl* and three *Wapl^fl/fl (Tg)Zp3*-Cre littermate females were analyzed. For Smc1α quantifications, two *Wapl^fl/fl* and two *Wapl^fl/fl (Tg)Zp3*-Cre littermate females were analyzed. For Scc1 quantifications, two *Wapl^fl/fl* and two *Wapl^fl/fl (Tg)Zp3*-Cre littermate females were analyzed. The total number of oocytes analyzed per condition is indicated in the figure. Fluorescence intensities are shown in a whisker plot graph indicating the median, first and third quartiles, and minimum and maximum values. P values are <0.0001 (Welch's *t* test). **(C)** Quantification of chiasma-like structures per bivalent in *Wapl^fl/fl* and *Wapl^Δ/Δ* oocytes. Error bars show SEM of three independent experiments (10 oocytes per condition, per experiment). Representative images of bivalent chromosome structure are shown, with Rec8 in gray, centromeres in red, and DNA in blue. The white arrowheads indicate the structures that we classified as chiasma-like structures. Scale bar, 5 μm.

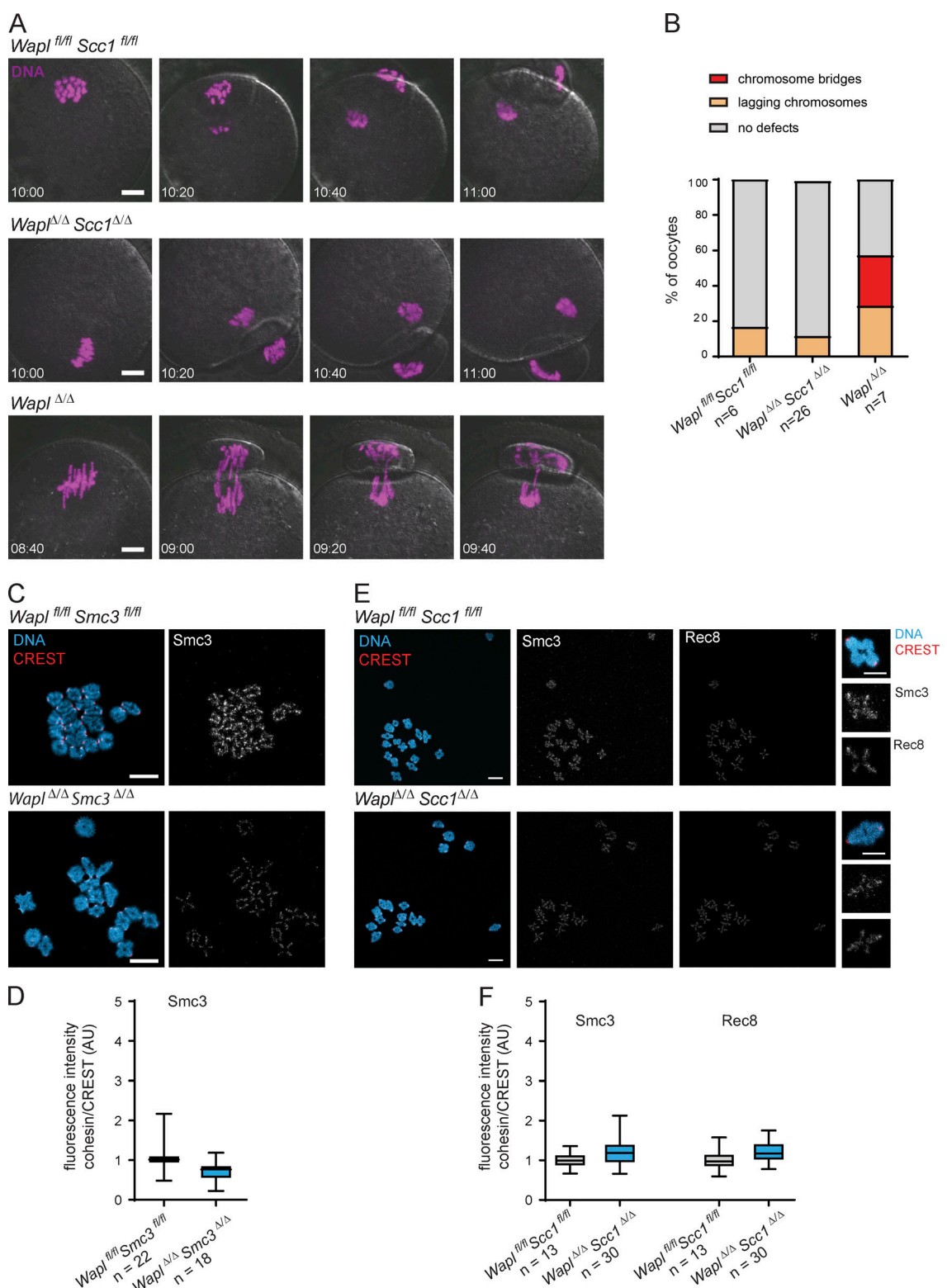

Figure 3. **The chromosome segregation defects observed in Wapl^Δ/Δ oocytes are a consequence of the increased chromosomal levels of Scc1-cohesin loaded onto chromatin long after DNA replication and meiotic recombination are completed. (A)** Representative stills of live-cell imaging videos showing chromosome segregation in Wapl^fl/fl^Scc1^fl/fl^, Wapl^Δ/Δ^Scc1^Δ/Δ^, and Wapl^Δ/Δ^ oocytes. DNA is shown in magenta. The time displayed indicates hours after GV breakdown. Scale is the same in all images; scale bar, 10 µm. **(B)** Quantification of chromosome segregation defects during meiosis I observed in live-cell imaging videos of Wapl^fl/fl^Scc1^fl/fl^, Wapl^Δ/Δ^Scc1^Δ/Δ^, and Wapl^Δ/Δ^ oocytes. Two Wapl^fl/fl^Scc1^fl/fl^, five Wapl^fl/fl^Scc1^fl/fl^ (Tg)Zp3-Cre, and two Wapl^fl/fl^ (Tg)Zp3-Cre females were analyzed. The total number of oocytes analyzed is indicated in the figure. The graph shows the percentage of oocytes for each chromosome-segregation phenotype (indicated in the legend). **(C)** Representative images of pro-metaphase I spreads of Wapl^fl/fl^Smc3^fl/fl^ and Wapl^Δ/Δ^Smc3^Δ/Δ^ oocytes. Centromeres are shown in red, Smc3 in gray, and DNA in blue. Scale bar, 10 µm. **(D)** Quantification of Smc3 fluorescence intensities per bivalent in relation to CREST

fluorescence intensity in *Wapl^(fl/fl)Smc3^(fl/fl)* and *Wapl^(Δ/Δ)Smc3^(Δ/Δ)* oocytes. Two *Wapl^(fl/fl)Smc3^(fl/fl)* and two *Wapl^(fl/fl)Smc3^(fl/fl) (Tg)Zp3-Cre* littermate females were analyzed. The total number of *Wapl^(fl/fl)Smc3^(fl/fl)* and *Wapl^(Δ/Δ)Smc3^(Δ/Δ)* oocytes analyzed is indicated in the figure. P value is <0.0001 (Welch's *t* test). **(E)** Representative images of pro-metaphase I spreads of *Wapl^(fl/fl)Scc1^(fl/fl)* and *Wapl^(Δ/Δ)Scc1^(Δ/Δ)* oocytes. Centromeres are shown in red, Smc3 and Rec8 in gray, and DNA in blue. Scale bar, 10 μm; inset scale bar, 5 μm. **(F)** Quantification of Smc3 and Rec8 fluorescence intensities per bivalent in relation to CREST fluorescence intensity in *Wapl^(fl/fl)Scc1^(fl/fl)* and *Wapl^(Δ/Δ)Scc1^(Δ/Δ)* oocytes. One *Wapl^(fl/fl)Scc1^(fl/fl)* and one *Wapl^(fl/fl)Scc1^(fl/fl) (Tg)Zp3-Cre* littermate females were analyzed. The total number of *Wapl^(fl/fl)Scc1^(fl/fl)* and *Wapl^(Δ/Δ)Scc1^(Δ/Δ)* oocytes analyzed is indicated in the figure. Fluorescence intensities are shown in a whisker plot graph as in Fig. 2 B). P values are <0.0001 (Welch's *t* test).

frequency was similar in fetal oocytes of *Wapl^(fl/fl)* and *Wapl^(fl/fl) (Tg)Zp3*-Cre embryos (Fig. S2, C and D). While chiasmata in *Wapl^(fl/fl)* oocytes lack Rec8 (Fig. 2 C), Rec8 was detectable at some chiasma-like structures in *Wapl^(Δ/Δ)* oocytes, implying that these were generated after recombination by a different mechanism. We hypothesize that Wapl loss leads to DNA breaks, possibly due to Scc1-cohesin accumulation affecting chromosome rigidity or loop extrusion activity, and these breaks are inefficiently repaired with a homologue-bias that leads to chromosome bridges (Fig. 1, C and D). Consistent with this, we found a significant increase in the number of phosphorylated histone H2AX (γH2AX) foci, a DNA damage marker, in *Wapl^(Δ/Δ)* compared with *Wapl^(fl/fl)* GV oocytes (before meiotic divisions; Fig. S2, E and F).

To test this notion further, we asked whether the chromosome segregation defects are due to chromosomal Scc1 accumulation. We injected *Wapl^(Δ/Δ) Scc1^(Δ/Δ)*, *Wapl^(Δ/Δ)*, and *Wapl^(fl/fl) Scc1^(fl/fl)* oocytes with mRNA encoding H2B-mCherry and performed live-cell imaging (Fig. 3, A and B; and Videos 4, 5, and 6). Lagging chromosomes in anaphase I were detected in all three genotypes. However, chromosome bridges were solely observed in *Wapl^(Δ/Δ)* and not in *Wapl^(Δ/Δ) Scc1^(Δ/Δ)* oocytes (Fig. 3, A and B; and Video 5), suggesting that bridges are caused by failure to release Scc1-cohesin from chromosomes. We conclude that the timely release of Scc1-cohesin by Wapl is important for meiotic chromosome segregation.

To distinguish whether Wapl releases cohesin that associated with chromosomes before *Wapl* deletion during oocyte growth or cohesin that is synthesized thereafter, we analyzed cohesin abundance in *Wapl^(Δ/Δ) Scc1^(Δ/Δ)* and *Wapl^(Δ/Δ) Smc3^(Δ/Δ)* oocytes. If Wapl releases newly synthesized cohesin, then lack of Scc1 or Smc3 expression should prevent accumulation of cohesin in double knockout oocytes. Alternatively, if Wapl releases cohesin that is associated with chromosomes before oocyte growth, then chromosomal cohesin would be expected to accumulate on double knockout oocytes. We found that Smc3 chromosomal abundance and localization are similar in *Wapl^(fl/fl) Scc1^(fl/fl)*, *Wapl^(fl/fl) Smc3^(fl/fl)*, *Wapl^(Δ/Δ) Scc1^(Δ/Δ)*, and *Wapl^(Δ/Δ) Smc3^(Δ/Δ)* oocytes (Fig. 3, C–F), suggesting that Wapl is releasing newly synthesized cohesin in growing oocytes.

## Wapl-mediated release of Scc1 regulates chromatin structure of oocytes

Since the chromosomal abundance of Scc1-cohesin affects chromosome segregation by a mechanism other than cohesion, we considered other functions for this complex in oocytes. Because Scc1-cohesin regulates higher-order chromatin organization and is essential for chromatin loops and TADs in somatic cells and fertilized eggs (zygotes; Gassler et al., 2017; Haarhuis and Rowland, 2017; Haarhuis et al., 2017; Rao et al., 2017; Schwarzer et al., 2017; Wutz et al., 2017; Davidson et al., 2019; Kim et al., 2019), we tested whether Scc1 is also essential for loops and TADs in oocytes. We performed single-nucleus Hi-C (snHi-C) on GV-stage (interphase) oocytes due to the technical requirement of manipulating nuclei (Flyamer et al., 2017; Gassler et al., 2018) and analyzed *Scc1^(fl/fl)* oocytes, in which Scc1 is detectable in the GV, and *Scc1^(Δ/Δ)* oocytes, in which Scc1 is undetectable (Fig. 4 A and Fig. S3 A). Contact enrichments were quantified using 12,000 de novo called loops from mouse embryonic fibroblast (MEF) bulk Hi-C data (unpublished data). We found contact enrichments in loops and TADs of *Scc1^(fl/fl)* oocytes, and these were undetectable in *Scc1^(Δ/Δ)* oocytes (Fig. 4 B and Fig. S3 B). Therefore, these chromatin structures depend on Scc1 in oocytes.

In somatic cells, increasing cohesin's chromatin residence time on chromosomes by Wapl depletion leads to cohesin accumulation in axial structures termed "vermicelli," which are thought to represent the base of chromatin loops (Tedeschi et al., 2013; Wutz et al., 2017; Haarhuis et al., 2017). Vermicelli were also observed by immunofluorescent staining of Scc1 and Smc3 in mature (surround nucleolus [SN]) and immature (non-SN [NSN]) *Wapl^(Δ/Δ)* oocytes, but not in *Wapl^(fl/fl)* oocytes (Fig. 4 A and Fig. S3 A). Polymer simulations and experimental data support the idea that increased residence time of cohesin, and thereby increased cohesin processivity, enables passage past barriers like CTCF (CCCTC-binding factor) and lead to extended loop extrusion, which results in an increase in contact enrichments at loop bases and larger loops (Gassler et al., 2017; Haarhuis et al., 2017; Wutz et al., 2017; Rao et al., 2017). We performed snHi-C to test whether Wapl prevents extended loop extrusion in oocytes. Unexpectedly, we found little or no difference in contact enrichment in average loops and TADs between *Wapl^(Δ/Δ)* and *Wapl^(fl/fl)* and no strong enrichment in longer average extruded loops in *Wapl^(Δ/Δ)* oocytes (Fig. 4, B and C; and Fig. S3, B and C). These results imply that vermicelli might simply reflect cohesin accumulation in axial structures. In addition, we observed a slight increase in compartmentalization in *Wapl^(Δ/Δ)* oocytes (Fig. 4 B and Fig. S3 B), which differs from the antagonism between loops and compartments in other cell types (Schwarzer et al., 2017; Haarhuis et al., 2017; Wutz et al., 2017; Rao et al., 2017; Gassler et al., 2017). The difference in oocytes could reflect cell type–specific chromatin features that are detected in compartment analysis.

We next considered how cohesin might accumulate in axial structures. It is conceivable that cohesin forms loops within loops and preventing its release enables complexes to translocate to loop anchors (Fig. 5 A). If so, then we would predict more complex loop-within-loop structures in the absence of Wapl and that these might be detected by a higher intra-loop contact

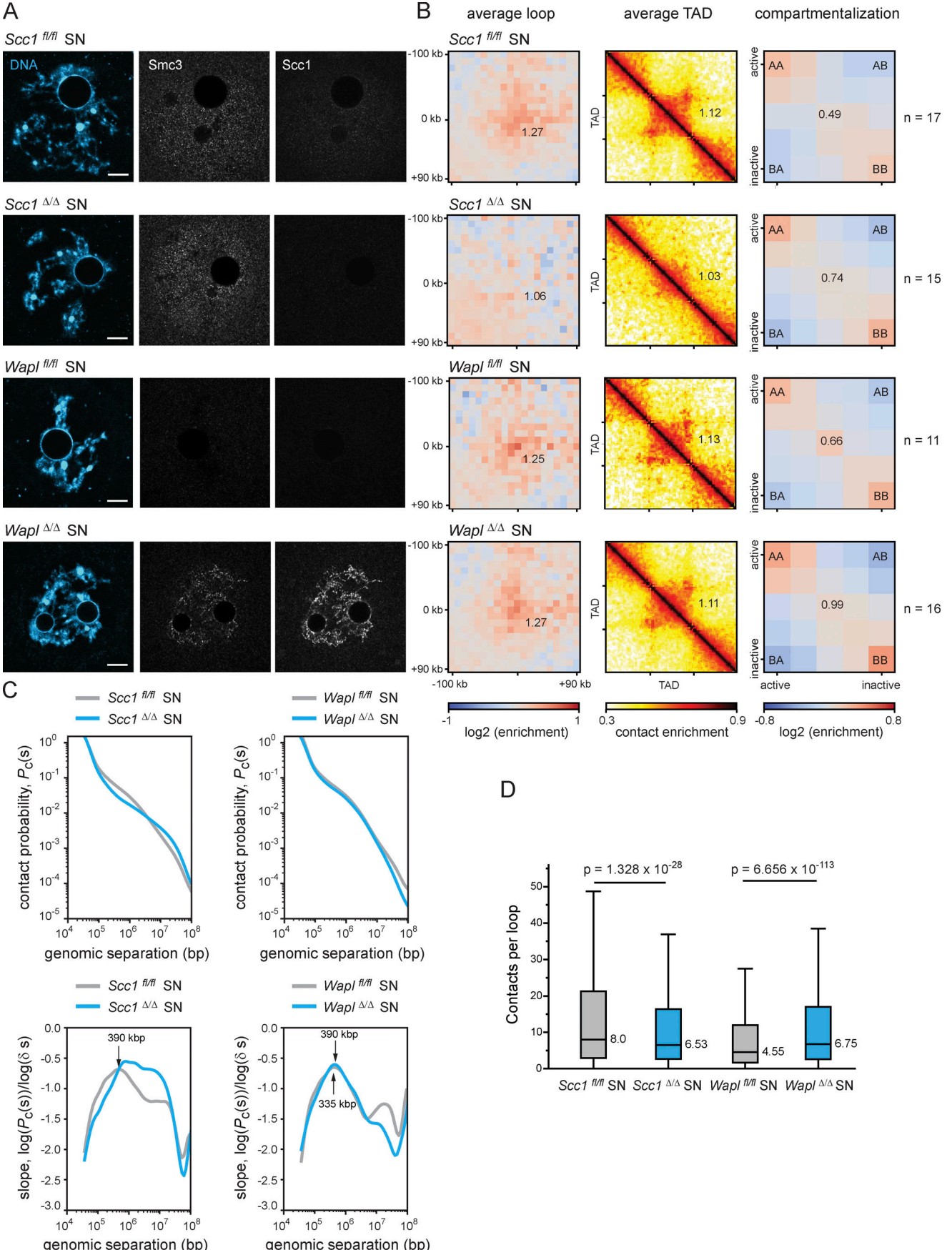

**Figure 4.** **DNA loops and TADs observed in mouse oocytes are largely dependent on Scc1-cohesin. (A)** Representative images of in situ fixed *Scc1^fl/fl^*, *Scc1^Δ/Δ^*, *Wapl^fl/fl^*, and *Wapl^Δ/Δ^* GV-oocytes in SN state (mature, SN). A single Z plane is shown to better visualize vermicelli structures in Wapl^Δ/Δ^ oocytes. DNA is shown in blue, and Smc3 and Scc1 in gray. Scale bar, 10 μm. **(B)** Average loops, TADs, and compartmentalization in *Scc1^fl/fl^*, *Scc1^Δ/Δ^*, *Wapl^fl/fl^*, and *Wapl^Δ/Δ^* GV-oocytes in SN state. The number of oocytes analyzed is indicated in the figure. Three *Wapl^fl/fl^*, four *Wapl^fl/fl^ (Tg)Zp3*-Cre, two *Scc1^fl/fl^*, and two *Scc1^fl/fl^ (Tg)Zp3*-Cre littermate females were analyzed. **(C)** $P_c(s)$ for *Scc1^fl/fl^*, *Scc1^Δ/Δ^*, *Wapl^fl/fl^*, and *Wapl^Δ/Δ^* GV-oocytes in SN state. Slopes of the $\log(P_c(s))$ curves for each condition are shown below the $P_c(s)$ plots. Gray lines show the controls *Scc1^fl/fl^* (left panel) and *Wapl^fl/fl^* (right panel), and blue lines show *Scc1^Δ/Δ^* (left panel) and *Wapl^Δ/Δ^* (right panel). **(D)** Quantification of the number of contacts within loop coordinates. This is calculated by extracting the contacts from the heat maps for each loop and normalizing by the sample size of each condition. The average number of contacts observed per loop is represented. The removal of Scc1-cohesin results in there being, on average, statistically less contacts within the loops, while in the absence of Wapl there are, on average, statistically more contacts per loop (paired Wilcoxon rank-sum test). The sample size is the same as in B.

frequency. We quantified contacts per loop in control, *Scc1^Δ/Δ^*, and *Wapl^Δ/Δ^* oocytes (Fig. 4 D and Fig. S3 D). Significantly fewer contacts were detected in *Scc1^Δ/Δ^* compared with *Scc1^fl/fl^* oocytes (Fig. 4 D). Contact frequency varied widely in *Scc1^fl/fl^* oocytes, consistent with loops being at different stages of extrusion. Importantly, significantly more contacts were detected in *Wapl^Δ/Δ^* compared with *Wapl^fl/fl^* oocytes, consistent with a predominance of loops within loops (Fig. 4 D and Fig. 5 A). We therefore propose that Wapl-mediated release of Scc1-cohesin regulates prophase I loops in oocytes.

If Scc1-cohesin is required for loops and TADs, then we would also expect to observe a loss of contacts over these genomic distances in *Scc1^Δ/Δ^* oocytes. We examined contact probability ($P_c(s)$) plots of the likelihood of contacts between pairs of genomic loci over increasing distance, and observed a loss of contacts >100 kbp to 1 Mbp and an increase in long-range contacts of >10 Mbp in *Scc1^Δ/Δ^* compared with *Scc1^fl/fl^* oocytes (Fig. 4 C and Fig. S3 C). The former likely reflects loss of loops and TADs, whereas the latter might reflect long-range polycomb interactions, which have recently been observed in other cell types (Donaldson-Collier et al., 2019; Wang et al., 2018; Rhodes et al., 2020; Ogiyama et al., 2018; Du et al., 2020). The first maximum of the slope derivative reflects the average loop size,

which increases from 390 kbp in *Scc1^fl/fl^* to 1.6 Mbp in *Scc1^Δ/Δ^* oocytes. The maxima in the absence of Scc1 are not well defined, presumably reflecting the stochastic nature of contacts without loop extrusion. Interestingly, the average loop size was much larger in *Scc1^Δ/Δ^* zygotes than in oocytes, namely >5 Mbp (Fig. 4 C and Fig. S3 C; Gassler et al., 2017). The difference could be either due to incomplete protein depletion or Scc1-independent chromatin structures in oocytes. Nevertheless, the findings that anchored loops became undetectable and average loop sizes expanded into the Mega-base pair range in *Scc1^Δ/Δ^* oocytes strongly support the conclusion that Scc1-cohesin is essential for normal loop structures in oocytes.

Last, we examined how $P_c(s)$ over genomic distance changes when Scc1 residence time is increased by Wapl loss. The $P_c(s)$ curves of *Wapl^fl/fl^* and *Wapl^Δ/Δ^* are similar, suggesting no gross changes in genome organization, and are consistent with lack of contact enrichment over loops and TADs. However, the slopes of *Wapl^Δ/Δ^* oocytes show that the standard deviation (broadness of the curve) of the average loop size is lower than that of *Wapl^fl/fl^* oocytes, which could be indicative of vermicelli formation (Fig. 4 C). The average loop sizes are 335 kbp and 390 kbp for *Wapl^fl/fl^* and *Wapl^Δ/Δ^* (Fig. 4 C), suggesting that increasing Scc1 residence time leads to some extended loop extrusion but

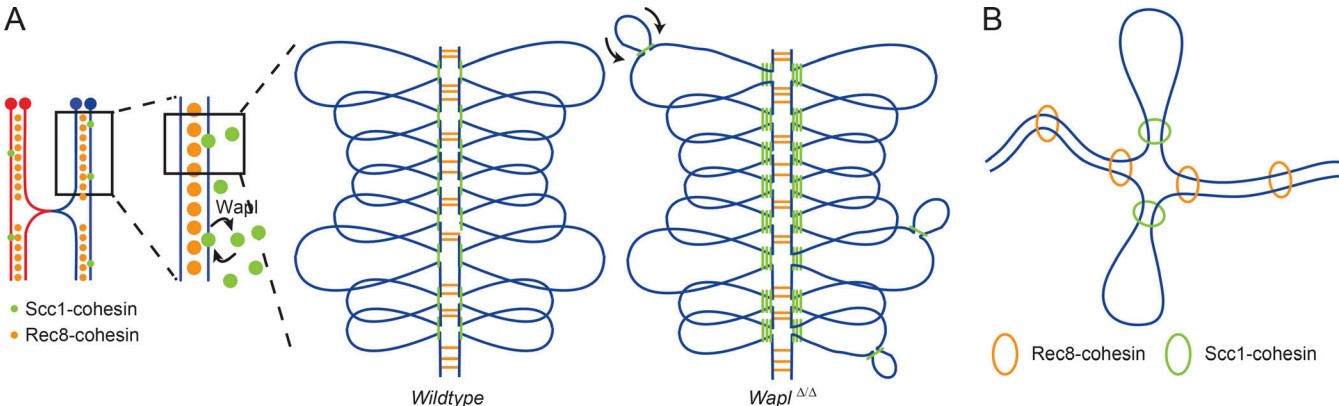

**Figure 5.** **Schematic representation illustrating the distinct chromosomal localization patterns of Scc1- and Rec8-cohesin in oocytes, and how these complexes contribute to two independent functions of cohesin. (A)** While Rec8-cohesin (orange) localizes between the two sister chromatids and is essential for chromosome cohesion, Scc1-cohesin (green) localizes onto the chromatid axis and contributes to loop and TAD formation. In wild-type oocytes, Wapl continuously removes Scc1-cohesin from chromatin, and in *Wapl^Δ/Δ^* oocytes, these cohesin complexes accumulate onto the chromatid axis. The differences in higher-order chromatin structure observed between *Wapl^fl/fl^* and *Wapl^Δ/Δ^* oocytes were very mild, possibly due to the existence of a physical constraint in oocytes that prevents the Scc1-cohesin–dependent loops to be extended in the absence of Wapl. We propose that in the absence of Wapl, the accumulation of Scc1-cohesin onto chromatin leads to extrusion of loops within loops, that when extruded to the maximum lead to accumulation of Scc1 onto the chromatid axis (vermicelli). **(B)** Chromosome organization and cohesion are mediated by distinct cohesin complexes in fully grown mouse oocytes: Rec8-cohesin (orange) mediates chromosome cohesion and Scc1-cohesin (green) mediates loop extrusion.

not as extensively as in other cell types (Gassler et al., 2017; Haarhuis et al., 2017; Schwarzer et al., 2017; Wutz et al., 2017). This implies that loop extrusion is somehow limited in oocytes. We speculate that loops have reached near maximal sizes, given that average HeLa cell loops are 262 kbp and increase to 387 kbp after Wapl depletion (Wutz et al., 2017). Alternatively, loop extrusion could be limited by additional barriers in oocytes, such as sister chromatid cohesion mediated by Rec8-cohesin (Chatzidaki, E., personal communication).

In summary, we demonstrate that Wapl is releasing Scc1-cohesin from chromosomes and is regulating 3D chromatin structure in meiosis I oocytes. We show that the timely release of Scc1-cohesin from bivalents, which are maintained by Rec8-cohesin, is important for proper chromosome segregation and production of euploid eggs. Whether the changes in chromatin structure due to Wapl loss are causally related to the chromosome segregation errors or whether these are two independent phenomena is unknown. Our data also do not allow us to exclude that a small fraction of Rec8-cohesin might also be released by Wapl. It has been proposed that Wapl releases cohesin in prophase I, but Rec8 levels were not analyzed in this study (Brieño-Enríquez et al., 2016). Instead, our work shows that the majority of Rec8 is resistant to the Wapl-mediated release pathway, either because Rec8-cohesin is not a good substrate for Wapl in oocytes, or because Rec8-cohesin is protected from Wapl by an unknown mechanism that specifically affects cohesive but not the noncohesive Scc1 complexes, analogous to how sororin protects cohesive cohesin in somatic cells (Nishiyama et al., 2010). In either case, our results explain why arm cohesion is maintained in the presence of Wapl until separase activation at the metaphase to anaphase I transition.

Specificity in Wapl's ability to release some but not other cohesin complexes from chromosomes has also been observed during meiotic recombination in *C. elegans*, where Wapl can release COH3/4-cohesin but not Rec8-cohesin from chromosomes (Crawley et al., 2016). COH3/4-cohesin has been proposed to be functionally related to Rad21L-cohesin, which only associates with chromosomes during meiotic recombination (Herrán et al., 2011; Ishiguro et al., 2011; Lee and Hirano, 2011; Severson and Meyer, 2014). Our work shows that during oocyte growth, i.e., long after recombination, Wapl preferentially releases Scc1-cohesin from mouse chromosomes.

Based on the observation that Wapl inactivation increases chromosomal COH3/4-cohesin levels and shortens the axes of pachytene chromosomes, it has further been speculated that different cohesin complexes may mediate chromosome organization and cohesion (Crawley et al., 2016). However, this hypothesis has not been tested as chromatin structure has never been directly analyzed in Wapl-depleted meiocytes in any species. Our observation that cohesion is generated by Rec8-cohesin (Tachibana-Konwalski et al., 2010) and loops are generated by Scc1-cohesin (this work) therefore shows for the first time that chromosome organization and cohesion are indeed mediated by distinct cohesin complexes (Fig. 5, A and B). This finding may also be relevant for somatic cells. Although these harbor cohesin with only one type of α-kleisin, different subunit compositions and post-translational modifications might similarly generate

distinct cohesin complexes that specialize in sister chromatid cohesion and loop extrusion.

## Materials and methods

### Mouse strains, husbandry, and genotyping
The mice used in this work were maintained and bred in accordance with the Austrian Animal Welfare law and with the guidelines of the international guiding principles for biomedical research involving animals (Council for International Organizations of Medical Sciences). Mice were kept at a daily cycle of 14-h light and 10-h dark with access to food ad libitum. All mice were bred in the IMBA animal facility. The number of mice used was kept as low as possible but in agreement with the standards used in the field. No statistical methods were used to estimate sample size. No randomization or blinding was used. $Scc1^{fl/fl}$ mice were bred on a mixed background (B6, 129, Sv; Ladstätter and Tachibana-Konwalski, 2016). $Wapl^{fl/fl}$ mice were bred on a primarily C57BL/6J background (Tedeschi et al., 2013). $Smc3^{fl/fl}$ mice were bred on a primarily C57BL/6J background (Busslinger et al., 2017). $Scc1^{fl/fl}$ $Wapl^{fl/fl}$ mice were bred on the same mixed background as $Scc1^{fl/fl}$ mice. $Smc3^{fl/fl}$ $Wapl^{fl/fl}$ mice were bred on the same mixed background as $Wapl^{fl/fl}$ mice. Experimental mice were obtained by mating of homozygous floxed females to homozygous floxed males carrying $Tg(Zp3-Cre)$ (Lewandoski et al., 1997).

For all experiments, with exception of Fig. S1, prophase I–arrested oocytes were harvested from 8–12-wk-old females.

### Oocyte culture and maturation
Ovaries were dissected from sexually mature female mice, which were sacrificed by cervical dislocation. Fully grown oocytes, naturally arrested in dictyate of prophase I, were isolated by physical disaggregation of the ovaries in M2 medium supplemented with the phosphodiesterase inhibitor 3-isobutyl-1-methylxanthine (IBMX; 200 µM in DMSO; Sigma-Aldrich) at 37°C. Mature oocytes were selected according to appearance (size, central nucleus, smooth zona pellucida), harvested with a mouth-pipette, and cultured in M16 media (Millipore, EmbryoMax) supplemented with IBMX at 37°C and 5% $CO_2$. Resumption of meiosis I was triggered by wash out of IBMX and successive culturing in M16 media. Only oocytes entering meiosis I within 90 min after IBMX release were used for the experiments. Oocyte cultivation was performed in ~40 µl drops covered with mineral oil (Sigma-Aldrich).

### Chromosome spreading of pro-metaphase I and metaphase II oocytes, immunofluorescence staining, and image acquisition
For pro-metaphase I and metaphase II spreads, oocytes were collected 4 h and 16 h after GV breakdown, respectively. Pro-metaphase I or metaphase II oocytes were washed into M2 droplets and then transferred through three droplets of Tyrode's solution until complete removal of the zona pellucida. Once the zona pellucida was removed, oocytes were washed through five droplets of M2 medium to avoid Tyrode's acid carry over. Oocytes were then incubated in an agarose plate containing FCS hypotonic solution (1:1 FCS and $H_2O$; Gibco) for 14 min at 37°C on

a heating plate. Oocytes were then collected in a multiwell slide and fixed overnight with fixation solution (1% PFA with 3 mM DTT and 0.15% Triton X-100) at RT in a humidified chamber. Slides were air dried at RT. Slides were washed two times with photoflo solution (0.08% in PBS; Kodak) for 5 min in a vertical shaker. These washes were followed by two more washes with PBS for 5 min with shacking. Finally, slides were washed two times with immunowashing solution (0.2% BSA and 0.1% Tween-20 in PBS) for 5 min. Oocytes were then incubated with blocking solution (10% goat serum, 2.5% BSA, and 0.1% Tween-20 in PBS) for 30 min at RT in a humidified dark box. After blocking, oocytes were incubated with primary antibodies for 1.5 h at RT. Immunofluorescent staining was performed using rabbit anti-Smc3 (Peters Lab ID A940), mouse anti-Scc1 (Millipore, 05-908), rabbit anti-Smc1α (Bethyl, A300-055A), mouse anti-Rec8 (gift from Yoshinori Watanabe, University of Sussex, Brighton, UK), and human CREST (gift from Arno Kromminga, Sonic Healthcare, Labor Lademannbogen, Hamburg, Germany) primary antibodies. Appropriate Alexa 488/568/647 conjugated secondary antibodies (Invitrogen) were used for visualization, and 5 µg/ml Hoechst 33342 was used for DNA counterstaining. Microscopy slides were prepared with ProlongGold mounting medium (Invitrogen). Samples were examined on a Zeiss LSM780 confocal microscope equipped with a 63×/1.4 plan-Apochromat oil differential interference contrast (DIC) objective lens using Zen Black software.

### Chromosome spreading of pachytene oocytes, immunofluorescence staining, and image acquisition
To obtain meiotic pachytene oocytes, embryonic ovaries of 17.5 d female embryos were isolated (Susiarjo et al., 2009). Ovaries were then incubated in a drop of hypotonic buffer (17 mM trisodium citrate-dihydrate, 50 mM sucrose, 5 mM EDTA, 0.5 mM DTT, and 30 mM Tris-HCl, pH 8.2; Sigma) for 25 min. Ovaries were disintegrated with 21 G needles to release cells in a sucrose drop (100 mM; Sigma). Cells were then fixed with 2% PFA containing 0.2% Triton X-100 overnight at RT in humidified box. Slides were air dried slowly. After air drying, pachytene oocytes were permeabilized with 0.1% Triton X-100 and were incubated with primary antibodies for 1.5 h at RT. Immunofluorescent staining was performed using rabbit anti-SYCP1 (Abcam, ab15090) and mouse anti-MLH1 (BD551092) primary antibodies. Appropriate Alexa 488/568 conjugated secondary antibodies (Invitrogen) were used for visualization and 1 µg/ml DAPI was applied for DNA counterstaining. Microscopy slides were prepared with Vectashield mounting medium. Imaging of spreads was performed on a Zeiss LSM780 confocal microscope equipped with a 40×/1.4 EC plan-Apochromat oil DIC objective lens using Zen Black software. For a more comprehensive protocol, see Silva et al. (2018).

### In situ fixation of GV oocytes, immunofluorescence staining, and image acquisition
For in situ staining, GV oocytes were fixed in 2% formaldehyde (in PBS) for 20 min at RT. Oocytes were post-extracted with permeabilization solution (0.3% BSA and 0.1% Triton X-100 in PBS). After permeabilization, oocytes were incubated for 1 h at

RT with blocking solution (10% goat serum, 0.3% BSA, and 0.1% Tween-20 in PBS). Oocytes were incubated with primary antibodies for 2 h at RT in a humidified dark box. After washing oocytes three times for 15 min in 0.2% Tween-20, oocytes were incubated with secondary antibodies for 1 h at RT in a humidified dark box. Excess of secondary antibodies was removed by washing oocytes three times for 15 min in 0.2% Tween-20 and DNA was stained with 10 µg/ml of Hoechst 33342 (Invitrogen). Microscopy slides were prepared with Vectashield mounting medium, and preparations were analyzed on a Zeiss LSM780 confocal microscope equipped with a 63×/1.4 plan-Apochromat oil DIC objective lens using Zen Black software.

For in situ staining presented in Fig. S2 E, GV oocytes were fixed in 4% PFA for 10 min at RT. Oocytes were post-extracted with 0.2% Triton X-100 in PBS. After permeabilization, oocytes were incubated for 1 h at RT with blocking solution (10% goat serum, 0.3% BSA, and 0.1% Tween-20 in PBS). Oocytes were incubated with primary antibodies overnight at 4°C in a humidified dark box. After washing oocytes three times for 15 min in 0.2% Tween-20, oocytes were incubated with secondary antibodies for 1 h at RT in a humidified dark box. Excess of secondary antibodies was removed by washing oocytes three times for 15 min in 0.2% Tween-20, and DNA was stained with 10 µg/ml of Hoechst 33342. Microscopy slides were prepared with Vectashield mounting medium, and preparations were analyzed on a Zeiss LSM700 confocal microscope equipped with a 63×/1.4 plan-Apochromat oil DIC objective lens using Zen Black software.

For the in situ stainings performed, we used the following primary antibodies: rabbit anti-Smc3 (Peters Lab ID A940), mouse anti-Scc1 (Millipore, 05-908), human CREST (gift from Arno Kromminga), and a mouse monoclonal anti-gamma H2A.X (Abcam, ab22551). Appropriate Alexa 488/568/647 conjugated secondary antibodies (Invitrogen) were used for visualization.

### In vitro culture, microinjection, and time-lapse confocal microscopy
Fully grown mouse GV oocytes were isolated and cultured as described in the oocyte culture and maturation section.

For low-resolution live-cell imaging, non-injected oocytes were imaged at 37°C and 5% $CO_2$ on an LSM800 confocal microscope equipped with a plan-Apochromat 20×/0.8 objective with 0.1% laser power using Zen Blue software. Image stacks of five slices of 7.5 µm were captured every 20 min.

For high-resolution live-cell imaging, GV oocytes were injected with mRNA for H2B-mCherry (0.5 pmol) and 2×EGFP-CenpC (2 pmol) to monitor chromosomes and centromeres, respectively. Microinjection of in vitro transcribed mRNA soluted in RNase-free water (mMessage mMachine T3 kit, Ambion) was performed in M2 media using a Pneumatic PicoPump (World Precision Instruments) and hydraulic micromanipulator (Narishige) mounted onto a Zeiss Axiovert 200 microscope equipped with a 10×/0.3 EC plan-neofluar and 40×/0.6 LD Apochromat objective. IBMX was washed out 2 h after injection to resume meiosis for the videos presented in Figs. 1 C and 3 A. Oocytes were imaged at 37°C and 5% $CO_2$ on a customized Zeiss LSM510 META confocal microscope equipped with a

C-Apochromat 63×/1.2 NA water immersion objective lens using AIM software. Chromosomes labeled with H2B-mCherry were tracked with an EMBL-developed tracking macro adapted to our microscope (Rabut and Ellenberg, 2004). Image stacks of 11 slices of 2 µm were captured every 20 min.

For the rescue experiment presented in Fig. S2, Wapl mRNA (2.3 pmol) and H2B-mCherry mRNA (0.5 pmol) were injected and IBMX was out 3 h after injection to resume meiosis, and chromosomes were spread and fixed 4 h after GV breakdown in pro-metaphase I as described above.

### Fluorescence intensity measurements in pro-metaphase I chromosome spreads

Cohesin fluorescence intensities were measured in maximum intensity projected images using ImageJ. Regions were drawn per bivalent based on Hoechst staining and cohesin, and CREST average fluorescence intensities were measured (mean gray value in ImageJ). Cohesin fluorescence intensities were normalized against CREST fluorescence intensities, taking in account antibody penetrance issues. For a more comprehensive protocol, see Silva et al. (2018).

### snHi-C

snHi-C was performed as previously described (Flyamer et al., 2017; Gassler et al., 2018). $Wapl^{fl/fl}$, $Wapl^{\Delta/\Delta}$, $Scc1^{fl/fl}$, and $Scc1^{\Delta/\Delta}$ oocytes were collected and fixed in 2% formaldehyde (Sigma) for 15 min at RT. DNA was stained with 0.2 µg/mL Hoechst 33342 and the maturation status of the oocytes (SN or NSN) was accessed using a Zeiss LSM780 confocal microscope equipped with a 63×/1.4 plan-Apochromat oil DIC objective lens. Oocytes were transferred to microwell plates (Sigma, M0815) and lysed on ice in lysis buffer (10 mM Tris-HCl, pH 8.0, 10 mM NaCl, 0.5% [v/v] NP-40 substitute [Sigma], 1% [v/v] Triton X-100, and 1× Halt Protease Inhibitor Cocktail [Thermo Scientific]) for at least 15 min. After a brief PBS wash, cells were transferred into a well containing 1× NEB3 buffer (New England BioLabs) with 0.6% SDS and incubated at 37°C for 2 h with shaking in a humidified atmosphere. The remaining cell nuclei were then washed in 1× DpnII buffer (New England BioLabs) plus 1× BSA (New England BioLabs) and further digested with DpnII (5 RE) at 37°C in a humidified atmosphere overnight. After a brief PBS and 1× ligation buffer wash, nuclei were transferred to 1× ligation buffer with 5U T4 ligase (Thermo Scientific) for 4.5 h at 16°C with slow shaking (50 rpm), and 30 min at RT. The nuclei were then transferred to 0.2 ml PCR tubes for the following steps. Whole-genome amplification was performed using illustra GenomiPhi v2 DNA amplification kit (GE Healthcare). In brief, nuclei were transferred to 3 µl sample buffer covered with mineral oil for decrosslinking overnight at 65°C. Nuclei were lysed by addition of 1.5 µl lysis solution (600 mM KOH, 10 mM EDTA, and 100 mM DTT) for 10 min at 30°C. After neutralization by addition 1.5 µl neutralizing solution (4 vol 1 M Tris HCl, pH 8.0 and 1 vol 3 M HCl), the whole genome amplification was performed by addition of 4 µl sample buffer, 9 µl reaction buffer, and 1 µl enzyme mixture. The samples were incubated at 30°C for 4 h followed by heat inactivation at 65°C for 10 min. High molecular weight DNA was purified using AMPure XP beads (Beckman Coulter, 1.8:1.0

beads:DNA ratio), and 1 µg was used to prepare Illumina libraries for sequencing (by VBCF NGS Unit) after sonicating to ~300–1,300 bp. The sonicated DNA was purified with a PCR purification kit (Qiagen) before library preparation (NEB Next Ultra II Library Prep kit for Illumina). Libraries were sequenced on HiSeq 2500 v4 with 125-bp paired-end reads (at VBCF NGS Unit); between 14 and 24 cells were sequenced per lane.

The Hi-C sequencing data was uploaded to GEO: https://www.ncbi.nlm.nih.gov/geo/query/acc.cgi?acc=GSE132264.

### Hi-C of MEFs

Hi-C libraries were generated as described in Wutz et al. (2017), with modifications as described below. $3 × 10^7$ MEFs were fixed in 2% formaldehyde for 10 min, after which the reaction was quenched with ice-cold glycine (0.125 M final concentration). Cells were collected by centrifugation (400 $g$ for 10 min at 4°C) and washed once with 50 ml PBS, pH 7.4. After another centrifugation step (400 $g$ for 10 min at 4°C), the supernatant was completely removed and the cell pellets were immediately frozen in liquid nitrogen and stored at −80°C. After thawing, the cell pellets were incubated in 50 ml ice-cold lysis buffer (10 mM Tris-HCl, pH 7.5, 10 mM NaCl, 5 mM MgCl$_2$, 0.1 mM EGTA, and 0.2% NP-40) for 1 h on ice. After centrifugation to pellet the cell nuclei (400 $g$ for 5 min at 4°C), nuclei were washed twice with 1.2× NEBuffer 2 (New England BioLabs) and transferred to 1.5 ml Eppendorf tubes. The nuclei were then collected by centrifugation step (400 $g$ for 5 min at 4°C) and a resuspended in 450 µl 1.2× NEBuffer 2 (New England BioLabs) with 13.5 µl of 20% SDS (0.6% final concentration), and the nuclei were incubated at 37°C for 2 h with agitation (900 rpm). Triton X-100 was added to a final concentration of 3.3%, and the nuclei were incubated at 37°C for 2 h with agitation (900 rpm). HindIII (New England BioLabs; 1,500 units per 7 million cells) restriction digestion was performed overnight at 37°C with agitation (900 rpm). Using biotin-14-dATP (Life Technologies), dCTP, dGTP, and dTTP (Life Technologies; all at a final concentration of 30 µM), the HindIII restriction sites were then filled in with Klenow (New England BioLabs) for 1 h at 37°C with shaking (700 rpm) for 10 s every 30 s. The nuclei were washed twice with ligation buffer and the ligation was performed for 12 h at 16°C (2,000 units T4 DNA ligase, Thermo Scientific) in a total volume of 100 µl ligation buffer (50 mM Tris-HCl, 10 mM MgCl$_2$, 1 mM ATP, 10 mM DTT, 100 µg/ml BSA, and 0.9% Triton X-100). After ligation, crosslinking was reversed by incubation with proteinase K (40 µl of 10 mg/ml in 300 µl Tris-EDTA buffer [TE]) at 65°C overnight. An additional proteinase K incubation (65 µl of 10 mg/ml per 7 million cells starting material) at 65°C for 2 h was followed by RNase A (Roche; 15 µl of 10 mg/ml per 7 million cells starting material) treatment and two sequential phenol/chloroform (Sigma) extractions. DNA precipitation was performed overnight at −20°C (3 M sodium acetate, pH 5.2 [1/10 volume] and ethanol [2.5 volumes]), and the DNA was then spun down (3,200 $g$ for 30 min at 4°C). The pellets were resuspended in 400 µl TE (10 mM Tris-HCl, pH 8.0 and 0.1 mM EDTA) and transferred to 1.5-ml Eppendorf tubes. After another phenol/chloroform (Sigma) extraction and DNA precipitation overnight at −20°C, the pellets were washed three times with 70% ethanol,

and the DNA concentration was determined using Quant-iT Pico Green (Life Technologies). To remove biotin from non-ligated fragment ends, 30–40 µg of Hi-C library DNA was incubated with T4 DNA polymerase (New England BioLabs) for 4 h at 20°C, followed by phenol/chloroform purification and DNA precipitation overnight at –20°C. After a wash with 70% ethanol, sonication was carried out to generate DNA fragments with a size peak around 400 bp (Covaris E220 settings: duty factor, 10%; peak incident power, 140 W; cycles per burst, 200; time, 55 s). After end repair (T4 DNA polymerase, T4 DNA polynucleotide kinase, Klenow [all New England BioLabs] in the presence of dNTPs in ligation buffer [New England BioLabs]) for 30 min at RT, the DNA was purified (Qiagen PCR purification kit). dATP was added with Klenow exo- (New England BioLabs) for 30 min at 37°C, after which the enzyme was heat inactivated (20 min at 65°C). A double-size selection using AMPure XP beads (Beckman Coulter) was performed: First, the ratio of AMPure XP beads solution volume to DNA sample volume was adjusted to 0.6:1. After incubation for 15 min at RT, the sample was transferred to a magnetic separator (DynaMag-2 magnet; Life Technologies), and the supernatant was transferred to a new Eppendorf tube, while the beads were discarded. The ratio of AMPure XP beads solution volume to DNA sample volume was then adjusted to 0.9:1 final. After incubation for 15 min at RT, the sample was transferred to a magnet (DynaMag-2 magnet; Life Technologies). Following two washes with 70% ethanol, the DNA was eluted in 100 µl of TLE (10 mM Tris-HCl, pH 8.0 and 0.1 mM EDTA). Biotinylated ligation products were isolated using MyOne Streptavidin C1 Dynabeads (Life Technologies) on a DynaMag-2 magnet (Life Technologies) in binding buffer (5 mM Tris, pH 8, 0.5 mM EDTA, and 1 M NaCl) for 30 min at RT. After two washes in binding buffer and one wash in ligation buffer (New England BioLabs), PE adapters (Illumina) were ligated onto Hi-C ligation products bound to streptavidin beads for 2 h at RT (T4 DNA ligase NEB, in ligation buffer, and slowly rotating). After washing twice with wash buffer (5 mM Tris, 0.5 mM EDTA, 1 M NaCl, and 0.05% Tween-20) and then once with binding buffer, the DNA-bound beads were resuspended in a final volume of 90 µl NEBuffer 2. Bead-bound Hi-C DNA was amplified with seven PCR amplification cycles (36–40 individual PCRs) using PE PCR 1.0 and PE PCR 2.0 primers (Illumina). After PCR amplification, the Hi-C libraries were purified with AMPure XP beads (Beckman Coulter). The concentration of the Hi-C libraries was determined by Bioanalyzer profiles (Agilent Technologies) and qPCR (Kapa Biosystems), and the Hi-C libraries were paired-end sequenced (HiSeqv4, Illumina) at VBCF NGS.

### Generation of genome-wide loop coordinates of MEFs

Hi-C libraries of MEFs were sequenced as described above. The Hi-C reads were processed using the HiCUP pipeline (Wingett et al., 2015). Di-tag mapping was performed using bowtie2 against the mm9 template, experimental artefacts, such as circularized reads and re-ligations, were filtered out, and duplicate reads were removed. Aligned Hi-C data were further processed with Juicer tools (Durand et al., 2016). Using binned Hi-C data, we computed Hi-C maps at various resolutions applying Knight–Ruiz balancing with the Juicer tools pre component.

Loop calling within these maps was performed on a dedicated GPU using the hiccups algorithm at 5-kb, 10-kb, and 25-kb resolutions and the default parameter values with a false discovery rate threshold of 0.1. The resulting loop sets were merged.

### Bioinformatics' analysis snHi-C

snHi-C data were processed and analyzed as described below (similarly as in Flyamer et al., 2017). The reads of each sample were mapped with bwa to mm9 and processed by the pairtools framework (https://pairtools.readthedocs.io/en/latest/) into pairs files, which was subsequently converted to COOL files, used as a container for Hi-C contact maps, by the cooler package (https://github.com/mirnylab/cooler).

We then analyzed loops by summing up snHi-C contact frequencies for loop coordinates of over 12,000 loops identified using the Hi-C data from wild-type MEFs published in Wutz et al., 2017. By averaging 20 × 20 matrices surrounding the loops and dividing the final result by similarly averaged control matrices, we removed the effects of distance dependence. For display and visual consistency with the loop strength quantification, we set the background levels of interaction to 1; the background is defined as the upper left 6 × 6 and lower right submatrices. For the quantification of loop strength, we divided the average signal in the middle 6 × 6 submatrix by the average signal in top left and bottom right (at the same distance from the main diagonal) 6 × 6 submatrices.

For average analysis of TADs, we used published TAD coordinates (Rao et al., 2014) for the CH12-LX mouse cell line. We averaged Hi-C maps of all TADs and their neighboring regions, chosen to be of the same length as the TAD, after rescaling each TAD to a 90 × 90 matrix. For visualization, the $P_c(s)$ of these matrices was rescaled to follow a shallow power law with distance (–0.25 scaling). TAD strength was quantified using $P_c(s)$ normalized snHi-C data. In Python notation, if M is the 90 × 90 TAD numpy array (where numpy is np) and L = 90 is the length of the matrix, then TAD_strength = box1/box2, where box1 = 0.5 * np.sum(M[0:L//3, L//3:2*L//3]) + 0.5 * np.sum(M[L//3:2*L//3,2*L//3:L]); and box2 = np.sum(M[L//3:2*L//3,L//3:2*L//3]).

Compartment saddle plot strength was quantified by the formula log(AA*BB/(AB*BA)), where AA, AB, BA, and BB represent the four corners of the iteratively corrected saddle plot matrix.

$P_c(s)$ curves were computed from 10-kb binned snHi-C data. We divided the linear genomic separations into logarithmic bins with a factor of 1.3. Data within these log-spaced bins (at distance, s) were averaged to produce the value of $P_c(s)$. Both $P_c(s)$ curves and their log-space slopes are shown following a Gaussian smoothing (using the scipy.ndimage.filters.gaussian_smoothing1d function with radius 0.8). Both the y-axis (i.e., log($P_c(s)$) and the x-axis (i.e., log[s]) were smoothed. The average loop size was determined by studying the derivative of the $P_c(s)$ curve in log–log space, that is, the slope of log($P_c(s)$). The location of the maximum of the derivative curve (i.e., position of the smallest slope) closely matches the average length of extruded loops.

To test the updated software version and the new loop coordinates (12,000 loops identified using the Hi-C data from

wild-type MEFs published in Wutz et al., 2017) we reanalyzed the data from Gassler et al. (2017). The numbers obtained are slightly different due to the use of 12,000 loops instead of the ~3,000 loops (Rao et al., 2014) and the few adjustments added to the analysis pipeline. We could reproduce the results published in Gassler et al. (2017), and we are therefore confident that our analysis is robust and trustworthy.

To determine the unanchored extruding loops, we extracted the submatrices of the contact matrix based of all loops coordinates of the over 12,000 MEFs loops. After masking the first two diagonals of the contact matrix we determined all contacts within each loop coordinate bound submatrix. For example, a loop can be anchored at position A and position B with A < B; however, we can also find contacts in the aggregate contact matrix that connect position A + n and B – m or A + m and B – n, etc. All such contacts were counted and normalized according to the sample size of the condition. For the statistical significance of the average number of contacts per loops between the conditions we used a paired Wilcoxon rank-sum test, which is a nonparametric test and therefore does not require the data to be normally distributed.

### Online supplemental material

Fig. S1 shows that Wapl depletion causes severe chromosome segregation defects in old oocytes. Fig. S2 shows that Wapl mRNA injection in GV oocytes prevents the increase in cohesin levels observed upon Wapl depletion. Fig. S3 shows that DNA loops and TADs observed in mouse oocytes are largely dependent on Scc1-cohesin. Video 1 shows that $Wapl^{fl/fl}$ oocytes display normal chromosome segregation during the first meiotic division. Video 2 shows that $Wapl^{\Delta/\Delta}$ oocytes present chromosome bridges and lagging chromosomes during the first meiotic division. Video 3 also shows that $Wapl^{\Delta/\Delta}$ oocytes present chromosome bridges and lagging chromosomes during the first meiotic division. Video 4 shows that $Wapl^{fl/fl} Scc1^{fl/fl}$ oocytes display normal chromosome segregation during the first meiotic division. Video 5 shows that $Wapl^{\Delta/\Delta} Scc1^{\Delta/\Delta}$ oocytes display normal chromosome segregation during the first meiotic division. Video 6 shows that $Wapl^{\Delta/\Delta}$ oocytes present chromosome bridges and lagging chromosomes during the first meiotic division.

## Acknowledgments

We are grateful to G. Wutz for MEF Hi-C data, Y. Watanabe for antibodies, H. Brandão (Harvard University, Cambridge, MA) for providing snHi-C software and advice, B. Dequeker for experimental assistance, and M. Schuh for hosting M.C.C. Silva for a few weeks in her laboratory. We thank K. Klien, M. Idarrago-Amado, K. Stecher, and H. Scheuch for technical assistance.

M.C.C. Silva was supported by European Molecular Biology Organization (EMBO: Long-Term Fellowship ALTF 1397-2012) and Human Frontier Science Program (HFSP: LT000759/2013-L). J. Gassler is supported by the European Research Council and the L'Oréal Austria Fellowship for Women in Science, and is an associated student of the DK Chromosome Dynamics (W1238-B20) supported by the Austrian Science Fund (FWF). This project was supported by a Wittgenstein award (Z196-B20) and by SFB Chromosome Dynamics (SFB-F3407-B19). Research in the laboratory of J.-M. Peters is supported by Boehringer Ingelheim, the Austrian Research Promotion Agency (Headquarter grant FFG-852936), the Austrian Science Fund (Wittgenstein Award Z196-B20 and SFB Chromosome Dynamics F3407-B19), the European Research Council under the European Union's Horizon 2020 research and innovation program (GA No 693949), and the Human Frontier Science Program (grant RGP0057/2018). Research in the laboratory of K. Tachibana is supported by the Austrian Academy of Sciences, a grant from the Austrian Science Fund (FWF) and the Herzfelder Foundation (P30613-B21), a European Research Council starting grant from the 7th Framework Program for Research (ERC-StG-336460), the Human Frontier Science Program (HFSP RGP0057/2018), and the Max Planck Society.

The authors declare no competing financial interests.

Author contributions: J.-M. Peters and K. Tachibana conceived the project and cosupervised M.C.C. Silva. Experiments were designed by M.C.C. Silva, S. Ladstätter, J. Gassler, and K. Tachibana. S. Ladstätter performed part of the live-cell imaging experiments in Fig. 1 C, S1 A, and 3 A. J. Gassler performed snHi-C in $Scc1^{fl/fl}$ and $Scc1^{\Delta/\Delta}$ oocytes. M.C.C. Silva performed all other experiments. S. Powell performed the Hi-C analysis. R. Stocsits performed the loop calling analysis of MEF Hi-C data. A. Tedeschi provided the $Wapl^{fl/fl}$ mouse. K. Tachibana, J.-M. Peters, and M.C.C. Silva wrote the manuscript with input from all authors.

Submitted: 16 June 2019

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

# Supplemental material

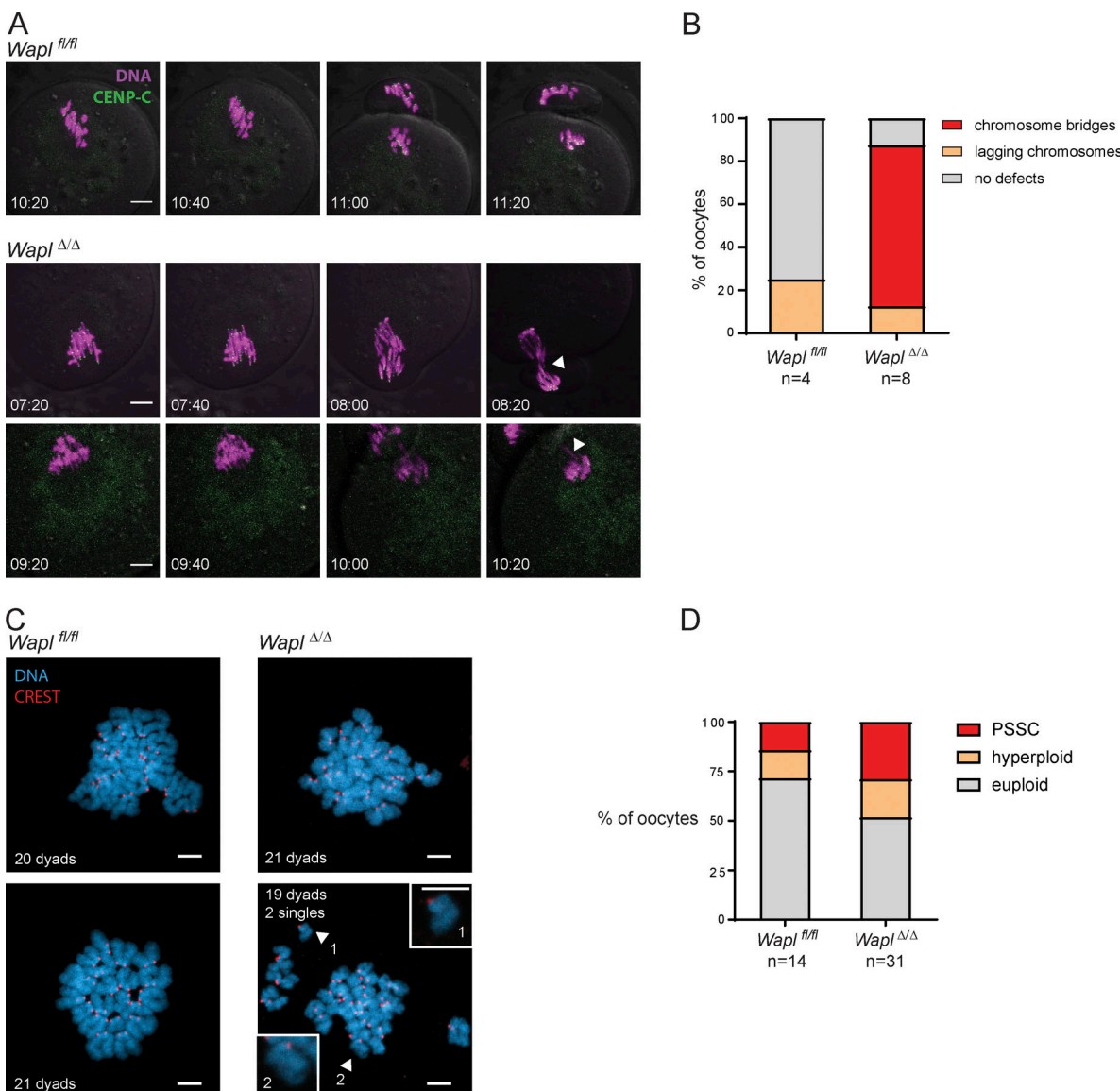

Figure S1. **Wapl depletion causes severe chromosome segregation defects in old oocytes. (A)** Representative stills of high-resolution live-cell imaging videos showing chromosome segregation in *Wapl*[fl/fl] and *Wapl*[Δ/Δ] oocytes isolated from 14-mo-old females. DNA is shown in magenta and centromeres (CENP-C) in green. White arrowheads indicate chromosome bridges, lagging chromosomes or misaligned chromosomes. The time displayed indicates hours after GV breakdown. Scale is the same in all images; scale bar, 10 μm. **(B)** Quantification of chromosome segregation defects observed in live-cell imaging videos of *Wapl*[fl/fl] and *Wapl*[Δ/Δ] oocytes isolated from 14-mo-old females. The number of oocytes analyzed is indicated in the figure. **(C)** Representative images of metaphase II spreads of *Wapl*[fl/fl] and *Wapl*[Δ/Δ] oocytes isolated from 14-mo-old females. Centromeres are shown in red and DNA in blue. The white arrowheads show single chromatids, which are highlighted in inset magnifications. Scale is the same in all images; scale bar, 5 μm. Inset scale bar, 5 μm. **(D)** The number of dyads was quantified in metaphase II chromosome spreads of *Wapl*[fl/fl] and *Wapl*[Δ/Δ] oocytes isolated from 14-mo-old females. The metaphase II chromosome spreads were classified in euploid (20 dyads), hyperploid (>20 dyads), and PSSCs. Four *Wapl*[fl/fl] and four *Wapl*[fl/fl] *(Tg)Zp3*-Cre littermate females were analyzed, and the total number of oocytes analyzed is indicated in the figure.

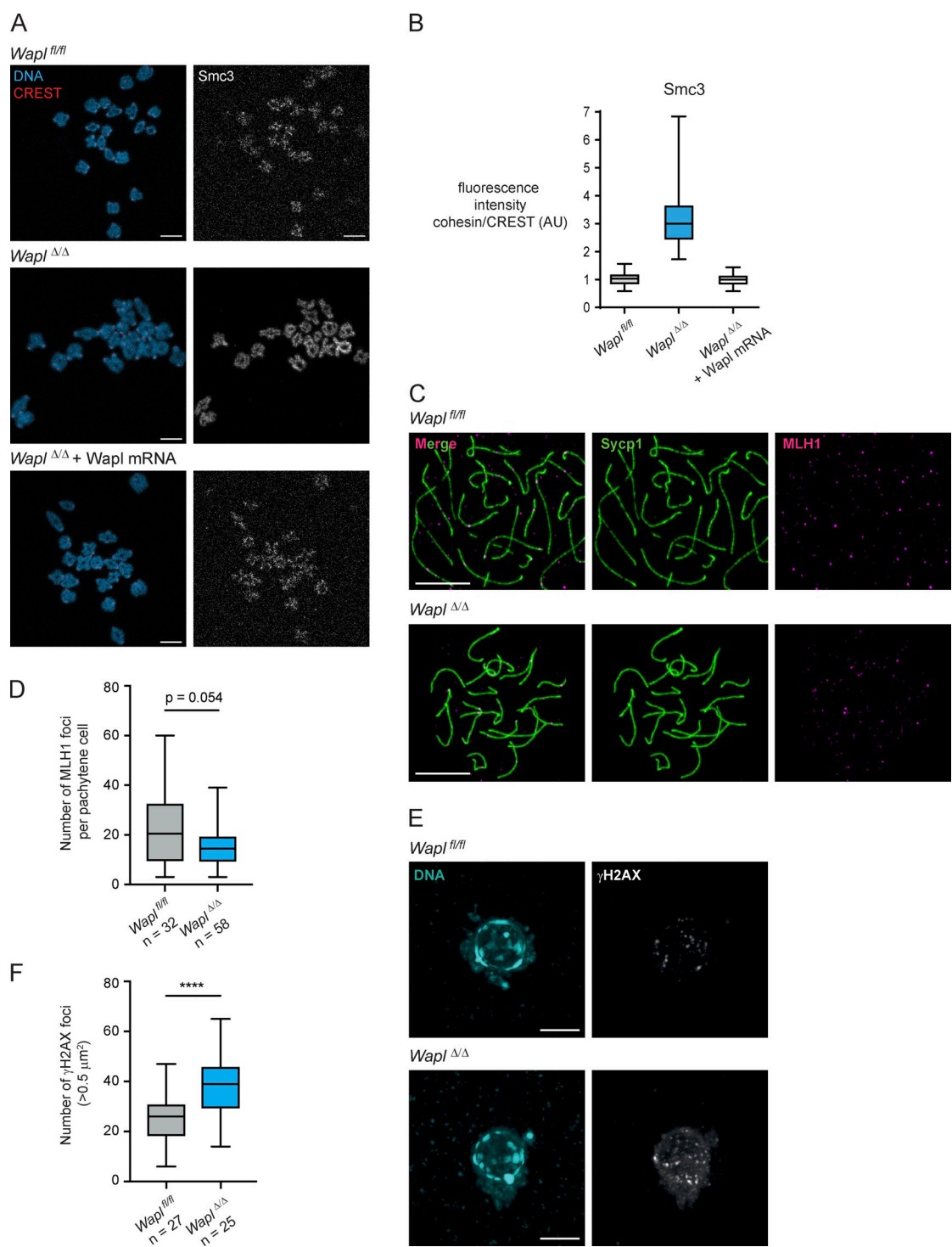

Figure S2. **Wapl mRNA injection in GV oocytes prevents the increase in cohesin levels observed upon Wapl depletion. Wapl depletion in oocytes using Zp3-Cre does not affect meiotic recombination, but leads to an increase in DNA breaks in mature oocytes. (A)** Representative images of pro-metaphase I chromosome spreads of *Wapl*[fl/fl], *Wapl*[Δ/Δ], and *Wapl*[Δ/Δ] oocytes injected with Wapl mRNA. Centromeres are shown in red, Smc3 in gray, and DNA in blue. Scale is the same in all images; scale bar, 10 µm. **(B)** Quantification of Smc3 fluorescence intensity per bivalent in relation to CREST fluorescence intensity in *Wapl*[fl/fl], *Wapl*[Δ/Δ], and *Wapl*[Δ/Δ] oocytes injected with Wapl mRNA. Approximately 10 oocytes were analyzed per condition. **(C)** Representative images of pachytene *Wapl*[fl/fl] and *Wapl*[Δ/Δ] oocytes isolated from 17.5 d *Wapl*[fl/fl] and *Wapl*[fl/fl] *(Tg)Zp3*-Cre female embryos. Chromosome spreads were stained with anti-MLH1 antibody to visualize recombination foci (magenta) and with anti-Sycp1 antibody to visualize synaptonemal complex (green). Scale is the same in all images; scale bar, 10 µm. **(D)** Number MLH1 foci was quantified in *Wapl*[fl/fl] and *Wapl*[Δ/Δ] oocytes. The number of pachytene oocytes analyzed is indicated in the figure. P value is 0.054 (Mann-Whitney test), indicating the difference observed is not significant. **(E)** Representative images of in situ fixed *Wapl*[fl/fl] and *Wapl*[Δ/Δ] GV-oocytes in SN state (mature, SN). DNA is shown in blue and γH2AX in gray. Scale is the same in all images; scale bar, 10 µm. **(F)** Number γH2AX foci was quantified in *Wapl*[fl/fl] and *Wapl*[Δ/Δ] oocytes. The number of oocytes analyzed is indicated in the figure. P value is <0.0001 (****, unpaired *t* test), indicating the difference observed is significant.

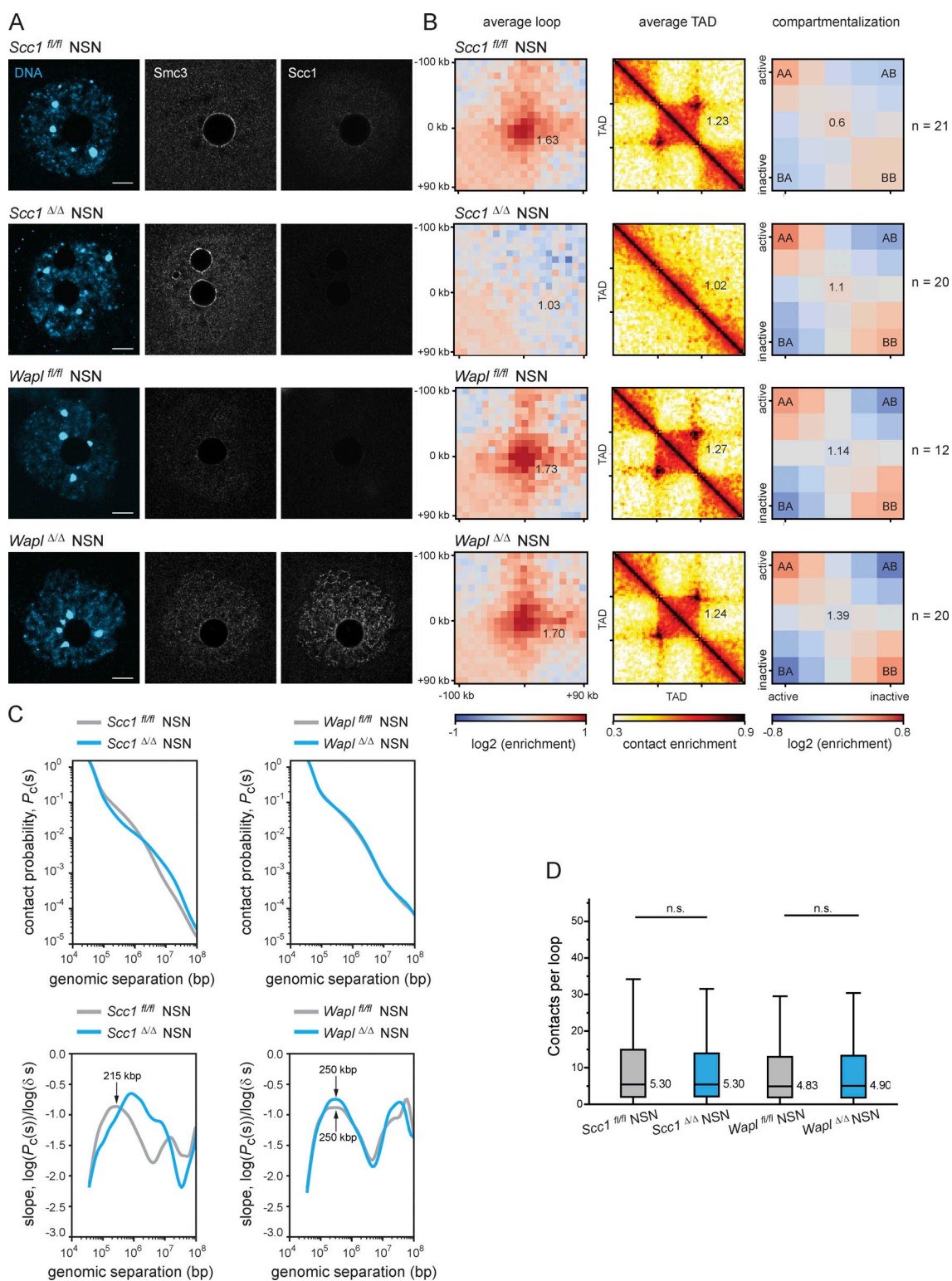

**Figure S3. DNA loops and TADs observed in mouse oocytes are largely dependent on Scc1-cohesin. (A)** Representative images of in situ fixed *Scc1*<sup>fl/fl</sup>, *Scc1*<sup>Δ/Δ</sup>, *Wapl*<sup>fl/fl</sup>, and *Wapl*<sup>Δ/Δ</sup> GV-oocytes in NSN state (immature, NSN). A single Z plane is shown to better visualize vermicelli structures in *Wapl*<sup>Δ/Δ</sup> oocytes. DNA is shown in blue, and Smc3 and Scc1 in gray. Scale is the same in all images; scale bar, is 10 μm. **(B)** Average loops, TADs, and compartmentalization in *Scc1*<sup>fl/fl</sup>, *Scc1*<sup>Δ/Δ</sup>, *Wapl*<sup>fl/fl</sup>, and *Wapl*<sup>Δ/Δ</sup> GV-oocytes in NSN state. The number of oocytes analyzed is indicated in the figure. Three *Wapl*<sup>fl/fl</sup>, four *Wapl*<sup>fl/fl</sup> *(Tg)Zp3*-Cre, two *Scc1*<sup>fl/fl</sup>, and two *Scc1*<sup>fl/fl</sup> *(Tg)Zp3*-Cre littermate females were analyzed. **(C)** $P_c(s)$ for *Scc1*<sup>fl/fl</sup>, *Scc1*<sup>Δ/Δ</sup>, *Wapl*<sup>fl/fl</sup>, and *Wapl*<sup>Δ/Δ</sup> GV-oocytes in NSN state. Slops of the log($P_c(s)$) curves for each condition are shown below the $P_c(s)$ plots. Gray lines show the controls *Scc1*<sup>fl/fl</sup> (left panel) and *Wapl*<sup>fl/fl</sup> (right panel), and blue lines show *Scc1*<sup>Δ/Δ</sup> (left panel) *and* Wapl<sup>Δ/Δ</sup> (right panel). **(D)** Quantification of the number of contacts within loop coordinates. The average number of contacts observed per loop is represented. Statistical significance was tested using a paired Wilcoxon rank-sum test (n.s., not significant). The number of oocytes analyzed is the same as in B.

Video 1. **$Wapl^{fl/fl}$ oocytes display normal chromosome segregation during the first meiotic division.** Related to Fig. 1 C. Representative live-cell imaging video of $Wapl^{fl/fl}$ oocytes expressing H2B-mCherry (magenta) and 2×EGFP-CENP-C (green). Images were captured every 20 min. Video speed is 5 frames/s.

Video 2. **$Wapl^{\Delta/\Delta}$ oocytes present chromosome bridges and lagging chromosomes during the first meiotic division.** Related to Fig. 1 C. Representative live-cell imaging video of $Wapl^{\Delta/\Delta}$ oocytes expressing H2B-mCherry (magenta) and 2×EGFP-CENP-C (green). Images were captured every 20 min. Video speed is 5 frames/s.

Video 3. **$Wapl^{\Delta/\Delta}$ oocytes present chromosome bridges and lagging chromosomes during the first meiotic division.** Related to Fig. 1 C. Additional representative live-cell imaging video of $Wapl^{\Delta/\Delta}$ oocytes expressing H2B-mCherry (magenta) and 2×EGFP-CENP-C (green). Images were captured every 20 min. Video speed is 5 frames/s.

Video 4. **$Wapl^{fl/fl} Scc1^{fl/fl}$ oocytes display normal chromosome segregation during the first meiotic division.** Related to Fig. 3 A. Representative live-cell imaging video of $Wapl^{fl/fl} Scc1^{fl/fl}$ oocytes expressing H2B-mCherry (magenta) and 2×EGFP-CENP-C (green). Images were captured every 20 min. Video speed is 5 frames/s.

Video 5. **$Wapl^{\Delta/\Delta} Scc1^{\Delta/\Delta}$ oocytes display normal chromosome segregation during the first meiotic division.** Related to Fig. 3 A. Representative live-cell imaging video of $Wapl^{\Delta/\Delta} Scc1^{\Delta/\Delta}$ oocytes expressing H2B-mCherry (magenta) and 2×EGFP-CENP-C (green). Images were acquired every 20 min. Video speed is 5 frames/s.

Video 6. **$Wapl^{\Delta/\Delta}$ oocytes present chromosome bridges and lagging chromosomes during the first meiotic division.** Related to Fig. 3 A. Representative live-cell imaging video of $Wapl^{\Delta/\Delta}$ oocytes expressing H2B-mCherry (magenta) and 2×EGFP-CENP-C (green). Images were acquired every 20 min. Video speed is 5 frames/s.

