## [Peer Review File · The Journal of Cell Biology]

Wapl releases Scc1-cohesin and regulates chromosome structure and segregation in mouse oocytes

Mariana Silva, Sean Powell, Sabrina Ladstätter, Johanna Gassler, Roman Stocsits, Antonio Tedeschi, Jan-Michael Peters, and Kikuë Tachibana

Corresponding Author(s): Kikuë Tachibana, Institute of Molecular Biotechnology of the Austrian Academy of Sciences (IMBA) and Jan-Michael Peters, IMP

Review Timeline:	Submission Date:	2019-06-16
	Editorial Decision:	2019-07-15
	Revision Received:	2019-11-22
	Editorial Decision:	2020-01-16
	Revision Received:	2020-01-29

Monitoring Editor: Arshad Desai

Scientific Editor: Tim Spencer

Transaction Report:

DOI: <https://doi.org/10.1083/jcb.201906100>

July 15, 2019

Re: JCB manuscript #201906100

Dr. Kikuë Tachibana
Institute of Molecular Biotechnology of the Austrian Academy of Sciences (IMBA)
Dr. Bohr Gasse 3
Vienna 1030
Austria

Dear Dr. Tachibana,

Thank you for submitting your manuscript entitled "Wapl releases Scc1-cohesin and regulates chromosome structure and segregation in mouse oocytes". Your manuscript has been assessed by expert reviewers, whose comments are appended below. While the reviewers find the work of interest, they have a number of comments and criticisms that will need to be addressed in a thorough revision and the revised manuscript will need to be re-evaluated by the reviewers before we can reach a decision on publication.

In our view, the reviewers have raised important and on-topic questions and provided a number of constructive comments to help improve the manuscript. We have discussed the comments editorially and believe the below should be the focus of new experimental efforts:

1) All 3 reviewers raise concerns about the chiasmata-like structures and the proposal that these are due to DNA double-strand breaks. We believe this point needs more effort to clarify.

2) Revs #2 and #3 indicate the importance of cell biological phenotypic analysis across different stages that is accompanied by quantification and better presentation/clarification (see General comment para #1 from Rev #2 and comments from Rev #3).

Please note also the comment from Rev #2 after the manuscript summary. This comment should be addressed in the revised discussion and an effort made throughout the manuscript to synthesize the different observations into a clear advance.

3) Rev #1 in point #8 brings up an interesting and informative experiment to help address the reason why Wapl loss does not lead to extended loops. If feasible, this experiment would extend and raise the impact of the work, but we will not require it in the revision.

All of the other points raised by the reviewers need to be addressed through text/figure changes. A reviewer response should accompany your revised manuscript and summarize all changes made in the revision.

Please let us know if you are able to address the major issues outlined above and wish to submit a revised manuscript to JCB. Note that a substantial amount of additional experimental data likely would be needed to satisfactorily address the concerns of the reviewers. It may be necessary to extend your manuscript to a full Research Article. Our typical timeframe for revisions is three to four months; if submitted within this timeframe, novelty will not be reassessed. We would be open to resubmission at a later date; however, please note that priority and novelty would be reassessed.

If you choose to revise and resubmit your manuscript, please also attend to the following editorial points. Please direct any editorial questions to the journal office.

GENERAL GUIDELINES:

Text limits: Character count for a Report is < 20,000; a full Research Article is < 40,000, not including spaces. Count includes title page, abstract, introduction, results, discussion, acknowledgments, and figure legends. Count does not include materials and methods, references, tables, or supplemental legends.

Figures: A Report may include up to 5 main text figures; a full Research Article may have up to 10 main text figures. To avoid delays in production, figures must be prepared according to the policies outlined in our Instructions to Authors, under Data Presentation, <http://jcb.rupress.org/site/misc/ifora.xhtml>. All figures in accepted manuscripts will be screened prior to publication.

*****IMPORTANT:** It is JCB policy that if requested, original data images must be made available. Failure to provide original images upon request will result in unavoidable delays in publication. Please ensure that you have access to all original microscopy and blot data images before submitting your revision.*******

Supplemental information: There are strict limits on the allowable amount of supplemental data. Reports may have up to 3 supplemental figures; a full Research Article may have up to 5 supplemental figures. Up to 10 supplemental videos or flash animations are allowed. A summary of all supplemental material should appear at the end of the Materials and methods section.

If you choose to resubmit, please include a cover letter addressing the reviewers' comments point by point. Please also highlight all changes in the text of the manuscript.

Regardless of how you choose to proceed, we hope that the comments below will prove constructive as your work progresses. We would be happy to discuss them further once you've had a chance to consider the points raised. You can contact the journal office with any questions, cellbio@rockefeller.edu or call (212) 327-8588.

Thank you for thinking of JCB as an appropriate place to publish your work.

Sincerely,

Arshad Desai, PhD
Editor, Journal of Cell Biology

Melina Casadio, PhD
Senior Scientific Editor, Journal of Cell Biology

Reviewer #1 (Comments to the Authors (Required)):

The manuscript by Silva et al. studies the effect of Wapl depletion on the segregation and conformation of chromosomes in mouse oocytes. The authors show that depletion of Wapl leads to

faster meiotic division and more missegregations. Meiotic cells contain two types of cohesin that differ in the kleisin subunit, and the authors show that Wapl depletion affects mainly, if not solely, cohesin that harbours the Scc1 kleisin variant. And finally, the authors show that Wapl depletion in oocytes remarkably has little or no effect on 3D genome organization. Overall the experiments are well performed, and the results are interesting. With some changes, I would consider this manuscript suitable for publication in JCB.

1) Figure 1e: Are these differences in chromosome numbers also observed by snHi-C? With this sequencing-based approach it may be possible to assess whether cells can lose chromosomes, have 2 of the same and zero of another. This could also reveal whether any specific chromosomes missegregate more often than others. And related to this, how often are multiple bridges observed?

2) Supplemental figure 1b is not clear to me. Is this a quantification of missegregations as in the examples shown in supplemental figure 1a? Or did they quantify the ploidy of the chromosomes.

3) Supplemental figure 1b: Are the missegregation observed in the fl/fl cells as severe as those in Wapl-depleted cells?

4) Figure 2 shows an increase of more than 5-fold for both Smc1 and Smc3, but only a 3-fold increase for Scc1. Is this difference due to the detection limit of Scc1? Or do the authors think that another pool of cohesin is also affected?

5) Figure 2c: The example pictures of the chiasmata are unclear to me. In many examples I cannot see the centromeric stainings. A zoom-in with a cartoon explaining what is what would also help.

6) Figure 4b shows a slight increase in compartmentalization for Wapl depleted cells? How do the authors reconcile this with the decrease in compartmentalization previously published for Wapl depletion in cells undergoing a mitotic cell cycle?

7) Figure 4d: The quantification in this boxplot is unclear to me. Is this the average of all loops? As the max of these diploid cells is 17, I would assume the maximum observed contacts is 34, but the plot runs up to 50. Or is this an aggregate analysis? Then it is unclear how this relates to the 12,000 loops that the data is compared to.

8) Figure 4: The observation that Wapl depletion does not lead to extended loops in oocytes is interesting. This result could indeed be due to cohesive Rec8 cohesin that blocks further loop enlargement. It would be powerful if the authors could back up this model with a Rec8 depletion experiment. Does Rec8 loss then permit loop extension upon Wapl depletion? If the authors could test this model with e.g. a TEV cleavable Rec8, this would be a great addition to the manuscript.

9) A point on semantics. Throughout the manuscript, the authors write about 'mitotic cells' when referring to cells undergoing a mitotic cell cycle. They here draw the contrast with cells undergoing a meiotic cell cycle. While this may be straightforward terminology for the meiosis crowd, this can be confusing for readers that don't think about meiotic vs mitotic cells on a daily basis. Most readers will think that 'mitotic' simply means that cells are not interphase but in M phase. I would therefore recommend that the authors refer to such cells as being 'non-meiotic' or just as 'cells undergoing a mitotic cell cycle'.

Minor points:

- Figure 2: The numbering of the y-axis of the second graph of 2b is off.

- Page 3 and page 10: The Rao et al., 2014 should be Rao et al., 2017.
- Page 9: Two key references are missing here. Rao et al., 2017 in the second sentence, and Haarhuis et al., 2017 at the top of the second paragraph.
- Page 11; The reference to the Haarhuis EMBO paper should be the Haarhuis Cell paper.
- Overall there are quite a few typo's. E.g. chromosome 'drigdes' in figure 3. and, 'single-cucleus' Hi-C in the methods.

Reviewer #2 (Comments to the Authors (Required)):

The manuscript by Silva et al describes the meiotic defects occurring upon conditional loss of Wapl in growing mouse oocytes, before entry into the first meiotic division. The authors show that Wapl loss results in an increase in Scc1 containing cohesin, but not Rec8 containing cohesin. Loss of Wapl leads to defects in meiosis I, but the phenotype is not severe, therefore female mice are not sterile. Nevertheless, chromosome bridges and an increase in aneuploidies after meiosis I is observed. The authors also observe an increase in chiasmata-like structures, which they say is probably due to increased DNA breaks due to loss of Wapl. The results indicate that Rec8 containing cohesin is not removed in a Wapl dependent manner in meiosis I. Scc1 is proposed to be required for TADS and chromatin loop organisation. The authors suggest that more Scc1 in Wapl knock-out oocytes leads to more complex chromatin loop structures, and therefore problems in chromosome segregation.

Nevertheless, in my opinion the main questions are not addressed:

Why should changes in chromatin loop structure lead to precocious sister chromatid segregation in meiosis I? Also, how more complex loop structures should result in chromosome bridges is not further explained. Overall, the take-home message is not clear to me: do we need to keep low levels of Scc1 during oocyte maturation to allow proper genome organisation and avoid the formation of vermicelli? But also, Scc1 is essential to loop and TAD formation. Is Wapl therefore responsible to keep a proper balance of Scc1 during oocyte maturation? Can we exclude that the phenotype observed is indeed due to Scc1 levels and not subtle changes in Rec8?

Major points:

General comments:

In the abstract the authors mention "residence time" of Scc1 on chromosomes, does that mean that Scc1 should be there only in immature GV oocyte and then unloaded by Wapl? A figure of immature and mature GV oocytes with Scc1 and Smc3 staining and quantifications is missing. Do we expect a decrease of Scc1 overtime? Also, in a Wapl deletion context, do we see a progressive accumulation of Scc1 and Smc3 over time comparing immature and mature oocytes? Scc1 presence and residence time has not been characterized in GV oocytes before. The results imply a mechanism promoting the loading of Scc1 cohesin subunit on chromosomes during GV growth. This is entirely new, has to be properly characterized, for a clear statement.

Are the extra loops, created after Wapl deletion, responsible for the chromosome stretching observed by live imaging? Does this modify Kt-Mt attachment? Is it possible to see if the modifications in genome organisation are homogenous (on all chromosomes) or more localised to some parts of the chromosomes (centromere, chiasmata, chromosome arms). How does the creation of extra loops relate to the phenotypes observed by live imaging?

1) The most surprising phenotype observed is the fact that more chiasmata are observed in Wapl knock-out oocytes. The knock-out event should take place after resolution of meiotic recombination events. Therefore, the authors conclude that these are not chiasmata that are due to meiotic cross-overs, but double strand breaks occurring upon increased Scc1 cohesin on chromosomes. This has to be verified, to ensure that the knock-out event didn't occur earlier than anticipated. Are the number of recombination events comparable in Wapl del/del oocytes, compared to Wapl fl/fl? Are there indeed more ds breaks observed in Wapl KO oocytes, after meiotic recombination has taken place, in growing GV oocytes?

The authors state that they don't see an enrichment of Rec8 on these structures but this is impossible to see on the image provided and no quantification is provided either (figure 2C).

2) Figure 3 is incomprehensible. Quantifications in b) do not correspond to strains shown in a), there is clearly much more Smc3 on chromosomes in the control (Wapl Smc3 fl/fl) than in the double KO a), but the quantification doesn't show a difference b).

3) In Figure 4d, contacts per loop are significantly different in Wapl KO compare to Wapl ctrl, unlike the Scc1 KO compared to ctrl. How can this allow the authors to conclude that Wapl deletion leads to loop reorganization due to Scc1? Shouldn't the Scc1 KO have a much stronger phenotype in this case, or at least as strong as the Wapl KO?

4) Figure 4a) Why is no Scc1 and Smc3 detected in the Wapl fl/fl? Shouldn't the Scc1 and Wapl fl/fl controls look the same?

5) The methodology of the HiC data is not well explained. How many oocytes are analysed, is there variation between different oocytes at the same GV stage. "Wutz and al., in preparation" is mentioned, is this a manuscript with the complete analysis of the HiC data? The authors need to provide way more information about how their HiC data were generated.

6) Figure S4 looks nearly the same as the one in a previous study from the lab (Gassler et al., 2017 EMBO, Figure 3B and EV3C). The authors state that the data were re-analysed but there are no new conclusions and the reader is just wondering why the numbers obtained are slightly different (which is not explained). Also showing data of both maternal and paternal zygote nucleus is not meaningful when the comparison is made for GV oocytes.

7) I am not sure how the comment that Wapl influences Rec8 residence time in a hypomorphic Rec8 mutant strain as data not shown, fits with the interpretation of Figure 2b, where the authors say that Rec8 is not or only a little affected by Wapl deletion (even though there is a difference in the quantifications shown). This data have to be included (and not mentioned as data not shown), because small differences in Rec8 containing cohesin may severely affect chromosome structure, too.

Genome reorganisation from immature to mature GV oocytes was well described in Flyamer IM et al., 2017, Nature. How does this fit with the role for Wapl and Scc1 proposed here?

Minor points:

Figure 1a: part of the scheme missing ? why 3 stars ?

1f: All sisters look separated

2b: Chromosome spread and quantification have to be on the same figure otherwise it is very confusing (add smc1 staining on this figure)
2c: Rec8 staining has to be magnified and quantified
3a: Smc3 staining is strong and doesn't reflect the quantification in 3b.
3b: Quantifications come from chromosome spreads from three different figures, this is impossible to follow (figure 2, figure 2S and figure 3).
4a-b: Have to be reversed, they appear in a different order in the text.
S2b: Image of wapl deletion and rescue have to be reversed to stay consistent with quantification.
S2d: not mentioned in the text
S3a-b: Have to be reversed, they appear in a different order in the text.
S3d: not mentioned in the text

Page 3: Reference Kudo et al: This paper doesn't show that Separase is required for cohesin cleavage in meiosis II, only in meiosis I

Reviewer #3 (Comments to the Authors (Required)):

Review of Silva et al., Wapl releases Scc1-cohesin and regulates chromosome structure.

This manuscript reports the consequences of Wapl deletion on chromosome structure and behavior in mouse oocytes. The authors use a Crelox system delete the Wapl gene and examine chromosome spreads or behavior in oocytes labeled with mCherry histone.

The effect of Wapl loss on meiosis I was very minor (about a 6% difference, 30 min over 8 hours). The authors state that Wapl depletion induced stretching (p.5). I did not see this in the figure or the movies. The chromosomes all seem to stretch in anaphase, and I couldn't find where the authors quantitated this behavior in the mutants. They report a large increase in chromosome bridges. Several examples are given (Fig. 1), but the authors do not report how these are quantitated. Are they seen only upon anaphase onset, is there a duration of time they must persist to be defined as a bridge, are they continuous, or is it just that the chromosomes are less compacted and therefore do not completely separate until the poles are further apart? It would be important to keep track of spindle lengths vs bridge formation.

Fig. 1f is missing labeling on the last two panels (and Inserts is spelled Insects).

There is concern about the heavy reliance on indirect immunofluorescence for quantitative measurements of protein level, as stated by the authors (p. 7). I would suggest a quantitative western at a minimum to confirm the results from the staining.

The authors see an increase in chiasma-like structures (Fig. 2). Unfortunately, they make the conclusion that loss of Wapl leads to DNA breaks that are inefficiently repaired. This is highly speculative with very little evidence. The authors then suggest that Wapl recognizes different pools of cohesin (existing, vs. newly synthesized). This was also very speculative, for which there is correlative data at best.

They make a point that Wapl must release Scc1-cohesin based on bridges in Figure 3. (bridges spelled dridges). The authors must define and quantify these bridges. I don't see how they can differentiate a bridge from a defect in compaction.

The HiC data is used for additional speculations about cohesin position at the base of specific loops.

In summary, this manuscript addresses the role of Wapl in cohesin function in meiosis. The data lack quantification and analysis in the case of bridges and needs corroboration in the case of getting protein quantification from indirect immunofluorescence. The models in Figure 5 are only one of

several that are consistent with the HiC data.

Dear Dr. Tachibana,

Thank you for submitting your manuscript entitled "Wapl releases Scc1-cohesin and regulates chromosome structure and segregation in mouse oocytes". Your manuscript has been assessed by expert reviewers, whose comments are appended below. While the reviewers find the work of interest, they have a number of comments and criticisms that will need to be addressed in a thorough revision and the revised manuscript will need to be re-evaluated by the reviewers before we can reach a decision on publication.

In our view, the reviewers have raised important and on-topic questions and provided a number of constructive comments to help improve the manuscript. We have discussed the comments editorially and believe the below should be the focus of new experimental efforts:

1) All 3 reviewers raise concerns about the chiasmata-like structures and the proposal that these are due to DNA double-strand breaks. We believe this point needs more effort to clarify.

2) Revs #2 and #3 indicate the importance of cell biological phenotypic analysis across different stages that is accompanied by quantification and better presentation/clarification (see General comment para #1 from Rev #2 and comments from Rev #3). Please note also the comment from Rev #2 after the manuscript summary. This comment should be addressed in the revised discussion and an effort made throughout the manuscript to synthesize the different observations into a clear advance.

3) Rev #1 in point #8 brings up an interesting and informative experiment to help address the reason why Wapl loss does not lead to extended loops. If feasible, this experiment would extend and raise the impact of the work, but we will not require it in the revision.

All of the other points raised by the reviewers need to be addressed through text/figure changes. A reviewer response should accompany your revised manuscript and summarize all changes made in the revision.

Please let us know if you are able to address the major issues outlined above and wish to submit a revised manuscript to JCB. Note that a substantial amount of additional experimental data likely would be needed to satisfactorily address the concerns of the reviewers. It may be necessary to extend your manuscript to a full Research Article. Our typical timeframe for revisions is three to four months; if submitted within this timeframe, novelty will not be reassessed. We would be open to resubmission at a later date; however, please note that priority and novelty would be reassessed.

If you choose to revise and resubmit your manuscript, please also attend to the following editorial points. Please direct any editorial questions to the journal office.

GENERAL GUIDELINES:

Text limits: Character count for a Report is < 20,000; a full Research Article is < 40,000, not including spaces. Count includes title page, abstract, introduction, results, discussion,

acknowledgments, and figure legends. Count does not include materials and methods, references, tables, or supplemental legends.

Figures: A Report may include up to 5 main text figures; a full Research Article may have up to 10 main text figures. To avoid delays in production, figures must be prepared according to the policies outlined in our Instructions to Authors, under Data Presentation, <http://jcb.rupress.org/site/misc/ifora.xhtml>. All figures in accepted manuscripts will be screened prior to publication.

IMPORTANT: It is JCB policy that if requested, original data images must be made available. Failure to provide original images upon request will result in unavoidable delays in publication. Please ensure that you have access to all original microscopy and blot data images before submitting your revision.

Supplemental information: There are strict limits on the allowable amount of supplemental data. Reports may have up to 3 supplemental figures; a full Research Article may have up to 5 supplemental figures. Up to 10 supplemental videos or flash animations are allowed. A summary of all supplemental material should appear at the end of the Materials and methods section.

If you choose to resubmit, please include a cover letter addressing the reviewers' comments point by point. Please also highlight all changes in the text of the manuscript.

Regardless of how you choose to proceed, we hope that the comments below will prove constructive as your work progresses. We would be happy to discuss them further once you've had a chance to consider the points raised. You can contact the journal office with any questions, cellbio@rockefeller.edu or call (212) 327-8588.

Thank you for thinking of JCB as an appropriate place to publish your work.

Sincerely,

Arshad Desai, PhD
Editor, Journal of Cell Biology

Melina Casadio, PhD
Senior Scientific Editor, Journal of Cell Biology

Reviewer #1 (Comments to the Authors (Required)):

The manuscript by Silva et al. studies the effect of Wapl depletion on the segregation and conformation of chromosomes in mouse oocytes. The authors show that depletion of Wapl leads to faster meiotic division and more missegregations. Meiotic cells contain two types of

cohesin that differ in the kleisin subunit, and the authors show that Wapl depletion affects mainly, if not solely, cohesin that harbours the Scc1 kleisin variant. And finally, the authors show that Wapl depletion in oocytes remarkably has little or no effect on 3D genome organization. Overall the experiments are well performed, and the results are interesting. With some changes, I would consider this manuscript suitable for publication in JCB.

1) Figure 1e: Are these differences in chromosome numbers also observed by snHi-C? With this sequencing-based approach it may be possible to assess whether cells can lose chromosomes, have 2 of the same and zero of another. This could also reveal whether any specific chromosomes missegregate more often than others. And related to this, how often are multiple bridges observed?

We thank the reviewer for their supportive comments and suggestions. It would indeed be interesting to use sequencing-based approaches to independently assess the aneuploidy status in these cells. In this case, the snHi-C method requires the manipulation and transfer of nuclei and therefore these experiments had to be carried out using germinal vesicle-stage (GV) oocytes, which are at a stage before meiotic resumption and the meiosis I division. We have also clarified this point in the manuscript. The correct stage to directly compare sequencing-based results and chromosome spreads (as in Fig 1E) would be metaphase II eggs, which unfortunately cannot be subjected to the snHi-C method due to lack of a nuclear envelope.

We observed chromosome bridges in 9 out of 21 *Wapl^{Δ/Δ}* oocytes, and 5 of those oocytes presented multiple chromosome bridges (Fig. 1C and D).

2) Supplemental figure 1b is not clear to me. Is this a quantification of missegregations as in the examples shown in supplemental figure 1a? Or did they quantify the ploidy of the chromosomes.

In the quantification shown in Fig. S1B we have quantified chromosome segregation defects observed in live-cell imaging movies. Representative still images of these movies are shown in Fig. S1A. We used the same classification of phenotypes as in Fig. 1D, namely “lagging chromosomes”, “chromosome bridges” and “no defects”. A schematic illustrating this chromosome segregation defects is presented in Review Fig. 1A.

3) Supplemental figure 1b: Are the missegregation observed in the fl/fl cells as severe as those in Wapl-depleted cells?

In both young and old oocytes, the chromosome missegregation defects observed in *Wapl^{Δ/Δ}* oocytes are much more severe, both in quantitative and in qualitative terms (mostly chromosome bridges), than the ones observed in *Wapl^{fl/fl}* oocytes. As shown in Fig. 1C and D, and in Fig. S1A and B, in *Wapl^{fl/fl}* oocytes we observed defects less frequently than in *Wapl^{Δ/Δ}* oocytes, and in *Wapl^{fl/fl}* oocytes we only observed lagging chromosomes, whereas in *Wapl^{Δ/Δ}* oocytes we also frequently observed chromosome bridges.

To test how objectively we were able to identify these phenotypes by visual inspection, we have now performed an additional image analysis. For this purpose, we have identified in each movie the first frame in which chromosome segregation in anaphase was visible. In these images we measured the fluorescence intensity of DNA in between the separating sets of chromosomes and the intensity of both chromosome sets, and calculated the ratio of these two values. This analysis, which is shown in Review Fig. 1B and C, confirmed our analysis by visual inspection.

4) Figure 2 shows an increase of more than 5-fold for both Smc1 and Smc3, but only a 3-fold increase for Scc1. Is this difference due to the detection limit of Scc1? Or do the authors think that another pool of cohesin is also affected?

Like the Reviewer, we were puzzled by this discrepancy and wondered whether Rec8-cohesin levels on chromosomes might also be increased by Wapl depletion. However, as is shown in Fig. 2B, we only observed a small increase, if any, of Rec8 signals on chromosomes after Wapl depletion. Furthermore, as is shown in Fig. 3E and F, we found that the strong increase in Smc3 levels observed in *Wapl^{ΔΔ}* oocytes (Fig. 2B) is almost entirely abrogated in *Wapl^{ΔΔ} Scc1^{ΔΔ}* oocytes (Fig. 3F). This indicates that the increase of chromosomal Smc1 and Smc3 after Wapl depletion must largely be caused by an increase in Scc1-cohesin, suggesting that Wapl predominantly releases Scc1-cohesin from chromosomes but no or only little Rec8-cohesin. We therefore suspect that the difference in increase of chromosomal Scc1 versus Smc1 α /Smc3 levels is predominantly caused by differences in the antibodies that we used.

An interesting related question is whether the differential effects of Wapl depletion on Scc1-cohesin and Rec8-cohesin reflect substrate specificity of Wapl for Scc1-cohesin, or whether the resistance of Rec8-cohesin to Wapl is a reflection of these complexes mediating sister chromatid cohesion, for which Scc1-cohesin is neither required nor sufficient in oocytes (Tachibana-Konwalski *et al.*, 2010). The finding from the Stemmann lab that Wapl can release Rec8-cohesin from chromatin when ectopically expressed in somatic cultured cells (Wolf *et al.*, 2018) indicates that the latter might be the case.

5) Figure 2c: The example pictures of the chiasmata are unclear to me. In many examples I cannot see the centromeric stainings. A zoom-in with a cartoon explaining what is what would also help.

To better visualize these chiasmata-like structures we added higher magnification images and showed the Rec8 channel in a separated image in Fig. 2C. We also indicated with white arrowheads in these images the structures that we classified as chiasma-like structures.

6) Figure 4b shows a slight increase in compartmentalization for Wapl depleted cells? How do the authors reconcile this with the decrease in compartmentalization previously published for Wapl depletion in cells undergoing a mitotic cell cycle?

The Reviewer is correct that this result is unexpected, and we have now mentioned it more

explicitly in the revised text on page 9. We presently do not know the reason for this phenomenon, which would be difficult to study as the molecular link between cohesin levels on chromatin and compartment strength is also not understood in somatic cells, and which was also not the focus of our current study.

7) Figure 4d: The quantification in this boxplot is unclear to me. Is this the average of all loops? As the max of these diploid cells is 17, I would assume the maximum observed contacts is 34, but the plot runs up to 50. Or is this an aggregate analysis? Then it is unclear how this relates to the 12,000 loops that the data is compared to.

For each condition, the average number of contacts within a loop was calculated according to the aggregate contact map for each of the 12,000 loop coordinates. For example, a loop can be anchored at position A and position B with $A < B$, however we could also find contacts in the aggregate contact map that connect position $A + n$ and $B - m$ or $A + m$ and $B - n$, etc. All such contacts were counted and normalized according to the sample size of the genotype analyzed.

We then performed a paired Wilcoxon rank-sum test to see if the average contact number of all 12,000 loops is statistically different between the conditions $Sccl^{fl/fl}$ vs $Sccl^{\Delta/\Delta}$ and $Wapl^{fl/fl}$ vs $Wapl^{\Delta/\Delta}$. In both cases, they are statistically different, as seen in the boxplot of Fig. 4D.

8) Figure 4: The observation that Wapl depletion does not lead to extended loops in oocytes is interesting. This result could indeed be due to cohesive Rec8 cohesin that blocks further loop enlargement. It would be powerful if the authors could back up this model with a Rec8 depletion experiment. Does Rec8 loss then permit loop extension upon Wapl depletion? If the authors could test this model with e.g. a TEV cleavable Rec8, this would be a great addition to the manuscript.

We thank the Reviewer for this suggestion. We have indeed tested this model, however, for strategic reasons decided to include the results from these experiments in a separate manuscript (Chatzidaki *et al.*, submitted). $Rec8^{TEV/TEV}$ $Wapl^{\Delta/\Delta}$ oocytes were isolated, microinjected with mRNA encoding TEV protease and subjected to snHi-C. We found that TEV protease expression indeed increased average loop sizes from 450 kbp to 515 kbp. These results support the hypothesis that cohesive cohesin can function as a barrier to loop extrusion, possibly explaining why we did not observe an increase in loop size in $Wapl^{\Delta/\Delta}$ oocytes. We attach the manuscript by Chatzidaki *et al.* for the Reviewers' information.

9) A point on semantics. Throughout the manuscript, the authors write about 'mitotic cells' when referring to cells undergoing a mitotic cell cycle. They here draw the contrast with cells undergoing a meiotic cell cycle. While this may be straightforward terminology for the meiosis crowd, this can be confusing for readers that don't think about meiotic vs mitotic cells on a daily basis. Most readers will think that 'mitotic' simply means that cells are not interphase but in M phase. I would therefore recommend that the authors refer to such cells as being 'non-meiotic' or just as 'cells undergoing a mitotic cell cycle'.

We thank the Reviewer also for this suggestion. We have now changed “mitotic cells” to “somatic cells”, throughout the text.

Minor points:

We apologize for these errors, which have now all been corrected.

- Figure 2: The numbering of the y-axis of the second graph of 2b is off.
This has been corrected.

- Page 3 and page 10: The Rao et al., 2014 should be Rao et al., 2017.
This has been corrected.

- Page 9: Two key references are missing here. Rao et al., 2017 in the second sentence, and Haarhuis et al., 2017 at the top of the second paragraph.
These references have now been added.

- Page 11; The reference to the Haarhuis EMBO paper should be the Haarhuis Cell paper.
This has been corrected.

- Overall there are quite a few typo's. E.g. chromosome 'drigdes' in figure 3. and, 'single-cucleus' Hi-C in the methods.
These have been corrected.

Reviewer #2 (Comments to the Authors (Required)):

The manuscript by Silva et al describes the meiotic defects occurring upon conditional loss of Wapl in growing mouse oocytes, before entry into the first meiotic division. The authors show that Wapl loss results in an increase in Scc1 containing cohesin, but not Rec8 containing cohesin. Loss of Wapl leads to defects in meiosis I, but the phenotype is not severe, therefore female mice are not sterile. Nevertheless, chromosome bridges and an increase in aneuploidies after meiosis I is observed. The authors also observe an increase in chiasmata-like structures, which they say is probably due to increased DNA breaks due to loss of Wapl. The results indicate that Rec8 containing cohesin is not removed in a Wapl dependent manner in meiosis I. Scc1 is proposed to be required for TADS and chromatin loop organisation. The authors suggest that more Scc1 in Wapl knock-out oocytes leads to more complex chromatin loop structures, and therefore problems in chromosome segregation.

Nevertheless, in my opinion the main questions are not addressed:

Why should changes in chromatin loop structure lead to precocious sister chromatid segregation in meiosis I? Also, how more complex loop structures should result in chromosome bridges is not further explained. Overall, the take-home message is not clear to me: do we need to keep low levels of Scc1 during oocyte maturation to allow proper genome organisation and avoid the formation of vermicelli? But also, Scc1 is essential to loop and TAD formation. Is Wapl therefore responsible to keep a proper balance of Scc1 during oocyte maturation? Can we exclude that the phenotype observed is indeed due to Scc1 levels and not subtle changes in Rec8?

We agree with the reviewer that it is not straightforward to explain how changes in chromatin organization relate to the chromosome segregation errors and resulting aneuploidies, and we emphasize that our aim was not to create the impression that we can draw a strong causal link based on our current knowledge. Indeed, it is not known how these are related in any cellular system and we can therefore only speculate. One possibility is that the phenomena of chromatin structure and chromosome segregation are unrelated but are both affected by cohesin dynamics. For example, efficient cleavage of Rec8-cohesin by separase might be impaired due to the presence of excessive chromosomal Scc1-cohesin and this could lead to anaphase bridges. Alternatively, the two are related if changes in loop structures, due to increased chromosomal residence time of Scc1, lead to more catenations and thus delay chromosome arm resolution. Regarding precocious sister centromere separation, there is now evidence from Adele Marston's lab that loop extrusion occurs at centromeres. An accumulation of Scc1-cohesin at the base of centromeric loops might displace pericentromeric Rec8-cohesin and weakening centromeric cohesion, and thus result in precocious sister chromatid segregation in meiosis I. Since these are speculations, we have summarized these only briefly on pages 4, 5 and 7.

Regarding the concern whether the phenotype is indeed due to Scc1 levels and not subtle changes in Rec8, our data does not allow us to exclude that a small fraction of Rec8-cohesin might also be released by Wapl. However, and importantly, we observe a full rescue of the chromosome segregation defects in *Wapl^{Δ/Δ} Scc1^{Δ/Δ}* oocytes (Fig. 3A and B). We think that this is strong evidence that the defects observed in *Wapl^{Δ/Δ}* oocytes are due to the increase in chromosomal levels of Scc1-containing cohesin.

Regarding the reviewer's question what the main message of our paper is, we would summarize this as follows: the timely release of Scc1-cohesin from bivalent chromosomes is important for proper chromosome segregation and for production of euploid eggs. In contrast, the majority of Rec8-cohesin is resistant to the Wapl-mediated release pathway and thus explains why arm cohesion is maintained until separase activation at the metaphase to anaphase I transition. We additionally show that Scc1-containing cohesin is essential for higher-order chromatin structure in meiosis I oocytes. Our previous finding that cohesion is mediated by Rec8-cohesin (Tachibana-Konwalski *et al.*, 2010) and our new observation that loops are dependent on Scc1-cohesin together show for the first time that chromosome organization and cohesion are mediated by distinct cohesin complexes during mammalian meiosis (Figs. 5A and B). This message may be relevant to all eukaryotic cells because, by extension, it suggests that differences in cohesin complexes either by subunit composition or post-translational modifications will alter the mode of chromosomal association and thus cohesin's function.

Major points:

General comments:

In the abstract the authors mention "residence time" of Scc1 on chromosomes, does that mean that Scc1 should be there only in immature GV oocyte and then unloaded by Wapl? A figure of immature and mature GV oocytes with Scc1 and Smc3 staining and quantifications

is missing. Do we expect a decrease of Scc1 overtime? Also, in a Wapl deletion context, do we see a progressive accumulation of Scc1 and Smc3 over time comparing immature and mature oocytes? Scc1 presence and residence time has not been characterized in GV oocytes before. The results imply a mechanism promoting the loading of Scc1 cohesin subunit on chromosomes during GV growth. This is entirely new, has to be properly characterized, for a clear statement.

Our comment on chromosomal “residence time” indeed implies a loading and a release reaction of Scc1, the latter of which is regulated by Wapl. We think that Scc1 is likely associating with chromosomes in a dynamic fashion but direct evidence for this would require FRAP experiments, which have not been established in oocytes. We do not expect that Scc1 should only be there in immature GV oocytes but also in mature GV oocytes. Indeed, we have followed the reviewer’s suggestion and performed Scc1 (Review Fig. 2A) and Smc3 quantifications (Review Fig. 2B) in SN and NSN GV oocytes (Review Fig. 2), based on the immunofluorescence staining shown in Fig. 4A and S3A. We found that there is a significant increase in Scc1 levels in both SN and NSN in *Wapl^{Δ/Δ}* oocytes compared with *Wapl^{fl/fl}* oocytes (Review Fig. 2A, p-value = 0,0054 (SN) and = 0,0498 (NSN) using an unpaired t-test). However, we did not observe a significant difference in Smc3 levels (Review Fig. 2B, p-value = 0,1067 (SN) and = 0,1847 (NSN) using an unpaired t-test), possibly because there is a pool of nucleoplasmic Smc3 associated with Rec8 that masks the relocalization of Smc3-Scc1 to vermicelli. The difference in overall levels of both Scc1 and Smc3 between SN and NSN oocytes is not significant (p-value = 0,4440 (Scc1) and = 0,4358 (Smc3) using an unpaired t-test). This result suggests that there is likely no progressive accumulation of Scc1 during the transition of immature to mature oocytes (Review Fig. 2), which could be due to depletion of soluble Scc1 as it all becomes chromosomally bound in the immature oocytes.

Are the extra loops, created after Wapl deletion, responsible for the chromosome stretching observed by live imaging? Does this modify Kt-Mt attachment? Is it possible to see if the modifications in genome organisation are homogenous (on all chromosomes) or more localised to some parts of the chromosomes (centromere, chiasmata, chromosome arms). How does the creation of extra loops relate to the phenotypes observed by live imaging?

Thank you for these good questions. We do not know if the extra loops are related to the chromosome stretching. We can speculate that the extra cohesin-mediated loops might limit condensin-mediated loops and that this somehow affects chromosome rigidity. It is not entirely clear how examining the kinetochore-microtubule attachments would provide insights into this and, whilst we would perform this experiment out of curiosity if we were studying tissue culture cells, we have in this case re-directed our efforts to use the genetically modified mice for other experiments, e.g. see the next point on meiotic recombination. Regarding the third part, the relatively low coverage of contacts per nucleus in snHi-C makes it very difficult to quantify and characterize the localization of specific modifications in higher order chromatin structure. Furthermore, it is not possible to map centromeres and chiasmata positions which vary from cell to cell.

1) The most surprising phenotype observed is the fact that more chiasmata are observed in *Wapl* knock-out oocytes. The knock-out event should take place after resolution of meiotic recombination events. Therefore, the authors conclude that these are not chiasmata that are due to meiotic cross-overs, but double strand breaks occurring upon increased *Scc1* cohesin on chromosomes. This has to be verified, to ensure that the knock-out event didn't occur earlier than anticipated. Are the number of recombination events comparable in *Wapl* del/del oocytes, compared to *Wapl* fl/fl? Are there indeed more ds breaks observed in *Wapl* KO oocytes, after meiotic recombination has taken place, in growing GV oocytes?

The conditional *Wapl* knockout used in this study is mediated by *Zp3-Cre* recombinase that is expressed and active during oocyte growth occurring after meiotic recombination. We therefore do not expect that the deletion should have taken place earlier, such as during meiotic recombination. Nevertheless, we agree with the reviewer that it would be good to know whether recombination events are comparable in oocytes isolated from *Wapl^{fl/fl}* and *Wapl^{fl/fl} (Tg)Zp3-Cre* females. We therefore isolated fetal oocytes at e17.5 from mice of both genotypes, performed pachytene spreads and immunostaining for *Mlh1* as a marker of crossovers. We observed similar numbers of meiotic crossover events in *Wapl^{fl/fl}* and *Wapl^{ΔΔ}* fetal pachytene oocytes, which have been included in manuscript in Fig. S2C and D. Therefore, we can exclude that the additional chiasma-like structures originated from additional crossover events during meiotic recombination.

In addition, we have tested whether there are more DNA breaks observed in *Wapl^{ΔΔ}* mature oocytes. We performed *in situ* fixation of oocytes from *Wapl^{fl/fl}* and *Wapl^{fl/fl} (Tg)Zp3-Cre* females and examined breaks using the DNA damage marker phosphorylated histone H2AX (γ H2AX). Interestingly, we observed a significant increase of γ H2AX foci number in mature *Wapl^{ΔΔ}* oocytes, which indicates persistence of DNA damage. We have included this data in Fig. S2E and F. Therefore, *Wapl* loss leads to formation of DNA breaks. We do not know the mechanism for the production of the DNA breaks but one speculation is that torsional stress due to loop extrusion might facilitate these. If the DNA breaks are inefficiently repaired with a homologue-bias, then these structures might manifest as chiasma-like structures and lead to chromosome bridges in meiosis I and aneuploidy.

The authors state that they don't see an enrichment of *Rec8* on these structures but this is impossible to see on the image provided and no quantification is provided either (Fig. 2C).

We are sorry for the poor image quality and for not properly explaining the *Rec8* enrichment, which we have both rectified in the revised manuscript. To better visualize these chiasma-like structures, we added higher magnification images and display *Rec8* immunostaining in a separate image. We also used white arrowheads to clearly indicate which structures were classified as chiasmata-like in Fig. 2C.

We would like to better explain our observations of *Rec8* localization to chiasma-like structures, which we take as a proxy that these were not generated by meiotic recombination. *Rec8* is largely undetectable on chiasma that are generated during meiotic recombination. Interestingly, we observed *Rec8* localization to the additional chiasma-like structures that appear only after *Wapl* depletion (Fig. 2C, white arrowhead). As requested by the reviewer,

we performed quantification of Rec8 in chiasma and in chiasma-like structures in different bivalent chromosomes of *Wapl^{ΔΔ}* oocytes (Review Fig. 3A). We observed that there is a significant increase in Rec8 levels in chiasma-like structures compared to natural chiasma (Review Fig. 3A; p-value = 0,0039 using an unpaired t-test). The natural chiasma structures show relatively high levels of Rec8, and this can be explained by the fact that the region designed around these chiasma structures using image J is slightly bigger than chiasma structures. The ratio of Rec8 levels between chiasma-like structures and natural chiasma is, in most bivalents analyzed, higher than 1 (Review Fig. 3B). This indicates that there is an enrichment of Rec8 in chiasma-like structures, compared to natural chiasma structures.

2) Figure 3 is incomprehensible. Quantifications in b) do not correspond to strains shown in a), there is clearly much more Smc3 on chromosomes in the control (*Wapl Smc3 fl/fl*) than in the double KO a), but the quantification doesn't show a difference b).

We apologize for this and have simplified the figure. We now display only the quantifications that correspond to the images shown. The images in Fig. 3C (previous Fig. 3A) are quantified in Fig. 3D (previous Fig. 3B), and the images in Fig. 3E (previous Fig. 3C) are quantified in Fig. 3F (previous Fig. 3D).

It is correct that the image shown in Fig. 3C (previous Fig. 3A) shows more Smc3 on chromosomes in control than in the double knockout. The quantification in Fig. 3D (previous Fig. 3B) shows that the Smc3 signal has a high variance in the controls and therefore the apparent difference is not statistically significant. Moreover, the point of this figure is to show that the increase in Smc3 on chromosomes upon *Wapl* depletion (see Fig. 2A and B) depends on newly synthesized Smc3. Therefore, what matters most is that there is no increase in Smc3 abundance in the double knockout but that Smc3 levels are lower than or similar to controls.

3) In Figure 4d, contacts per loop are significantly different in *Wapl* KO compare to *Wapl* ctrl, unlike the *Sccl* KO compared to ctrl. How can this allow the authors to conclude that *Wapl* deletion leads to loop reorganization due to *Sccl*? Shouldn't the *Sccl* KO have a much stronger phenotype in this case, or at least as strong as the *Wapl* KO?

Our original statistical analysis with the t-test did show significance for both *Wapl^{fl/fl}* versus *Wapl^{ΔΔ}* and *Sccl^{fl/fl}* versus *Sccl^{ΔΔ}* oocytes, however we misinterpreted $p = 0$ as non-significant rather than $p < 2.23E-308$, which is due to the limitations of double precision floating point arithmetic. Nevertheless, we decided to change to the Wilcoxon test since it makes no distributional assumptions and has the advantage of providing exceptionally good robustness. Having said that, the t-test is also quite robust, especially for large sample sizes, just not to the same extent as the Wilcoxon-Mann-Whitney test. For large sample sizes like our 12,000 data points, the t-test does not require an assumption that the data is normally distributed, it only needs to come from a distribution to which the Central Limit Theorem applies. However, we have not tested this on our dataset.

4) Figure 4a) Why is no Scc1 and Smc3 detected in the Wapl fl/fl? Shouldn't the Scc1 and Wapl fl/fl controls look the same?

We have previously shown that Scc1 is below the detection threshold in control oocytes (Tachibana-Konwalski *et al.*, 2010). The signals for Scc1 and Smc3 are both very low in control strains and can vary slightly between strains of different genetic backgrounds. For this reason, we always compare strains of knockout versus flox, rather than wild-type.

5) The methodology of the HiC data is not well explained. How many oocytes are analysed, is there variation between different oocytes at the same GV stage. "Wutz and al., in preparation" is mentioned, is this a manuscript with the complete analysis of the HiC data? The authors need to provide way more information about how their HiC data were generated.

We apologize if we did not clearly explained the HiC methodology. The number of oocytes analyzed by snHi-C per condition is indicated in Fig. 4B and S3B. A detailed protocol of the snHi-C methodology has been published recently in Gassler *et al.* 2018, as mentioned in the material and methods section. We also provide information in the methods section on how this samples were processed. The Wutz *et al.* only provides the 12000 loop coordinates observed in MEFs.

6) Figure S4 looks nearly the same as the one in a previous study from the lab (Gassler et al., 2017 EMBO, Figure 3B and EV3C). The authors state that the data were re-analysed but there are no new conclusions and the reader is just wondering why the numbers obtained are slightly different (which is not explained). Also showing data of both maternal and paternal zygote nucleus is not meaningful when the comparison is made for GV oocytes.

We originally included Fig. S4 to show our due diligence analysis in testing the updated software version on previously analyzed data from Gassler *et al.* 2017. The numbers obtained are slightly different due to the use of 12,000 loops (Wutz *et al. in preparation*) instead of the ca. 3,000 loops (Rao *et al.* 2014) and some adjustments to the pipeline. Since we could reproduce the results published in Gassler *et al.* 2017, we are confident that our analysis is robust and trustworthy.

Due to space constraints and the risk of confusion, we have decided to remove this supplementary figure. Instead, we have included a statement in the methods section of how the updated software version was validated.

7) I am not sure how the comment that Wapl influences Rec8 residence time in a hypomorphic Rec8 mutant strain as data not shown, fits with the interpretation of Figure 2b, where the authors say that Rec8 is not or only a little affected by Wapl deletion (even though there is a difference in the quantifications shown). This data have to be included (and not mentioned as data not shown), because small differences in Rec8 containing cohesin may severely affect chromosome structure, too.

We removed this sentence, to avoid confusion and because it does not add significant value to the message of the current manuscript. Moreover, we still do not fully understand the detailed molecular mechanism underlying the phenotype observed in our hypomorphic Rec8 mutant.

Genome reorganisation from immature to mature GV oocytes was well described in Flyamer *et al.*, 2017, Nature. How does this fit with the role for Wapl and Scc1 proposed here?

We appreciate the recognition of our previous work but do not fully understand the issue at hand. In Flyamer *et al.* 2017, we examined wild-type immature and mature GV oocytes and found that their genomes are organized into loops, TADs and compartments. We did not perform any genetic manipulations to test the proteins required for generating these higher-order chromatin structures. In contrast, our new work focuses on the mechanisms that generate these structures and we test whether there is a division of labor between cohesive and loop extruding cohesin complexes. This manuscript shows that Scc1 is essential for loops and TADs in oocytes, which was not known before. Moreover, we show that Wapl is releasing Scc1-cohesin from chromosomes of oocytes and preventing this timely release leads to vermicelli formation, errors in chromosome segregation and aneuploidy. Therefore, this manuscript provides insights into molecular players that generate higher-order chromatin structures in oocytes.

Minor points:

Figure 1a: part of the scheme missing ? why 3 stars ?

We have now explained it better in figure legend 1A.

The 3 stars represent the different cycles of oocyte growth that precede each round of meiotic divisions.

1f: All sisters look separated

The metaphase II chromosome spreads look similar to chromosome spreads observed in previously published studies such as in Chiang *et al.* 2010 (with the majority of sister kinetochores separated).

2b: Chromosome spread and quantification have to be on the same figure otherwise it is very confusing (add smc1 staining on this figure)

We agree and have rearranged the figure accordingly.

2c: Rec8 staining has to be magnified and quantified

To better visualize these chiasmata-like structures, we added higher magnification images and showed Rec8 immunostaining in a separated image (Fig. 2C). We also used white arrowheads to indicate chiasma-like structures.

3a: Smc3 staining is strong and doesn't reflect the quantification in 3b.

We have explained this point above to answer to point 2 of reviewer #2.

3b: Quantifications come from chromosome spreads from three different figures, this is impossible to follow (figure 2, figure 2S and figure 3).

We have simplified this and explained it above.

4a-b: Have to be reversed, they appear in a different order in the text.

We have reversed the order in the text and kept the figure order.

S2b: Image of wapl deletion and rescue have to be reversed to stay consistent with quantification.

This has been corrected.

S2d: not mentioned in the text

We have removed this panel since it did not provide useful information.

S3a-b: Have to be reversed, they appear in a different order in the text.

We have reversed the order in the text and kept the figure order.

S3d: not mentioned in the text

We have corrected this.

Page 3: Reference Kudo et al: This paper doesn't show that Separase is required for cohesin cleavage in meiosis II, only in meiosis I

We have corrected this and added Tachibana-Konwalski *et al.* 2010.

Reviewer #3 (Comments to the Authors (Required)):

Review of Silva et al., Wapl releases Scc1-cohesin and regulates chromosome structure.

This manuscript reports the consequences of Wapl deletion on chromosome structure and behavior in mouse oocytes. The authors use a Crelox system delete the Wapl gene and examine chromosome spreads or behavior in oocytes labeled with mCherry histone. The effect of Wapl loss on meiosis I was very minor (about a 6% difference, 30 min over 8 hours). The authors state that Wapl depletion induced stretching (p.5). I did not see this in the figure or the movies.

We would like to refer you to Fig. 3E (last panel) where it is more clearly visible.

The chromosomes all seem to stretch in anaphase, and I couldn't find where the authors quantitated this behavior in the mutants. They report a large increase in chromosome bridges. Several examples are given (Fig. 1), but the authors do not report how these are quantitated. Are they seen only upon anaphase onset, is there a duration of time they must persist to be defined as a bridge, are they continuous, or is it just that the chromosomes are less compacted and therefore do not completely separate until the poles are further apart? It would be important to keep track of spindle lengths vs bridge formation.

Our current classification of lagging chromosomes versus anaphase bridges is based on morphological criteria. Lagging chromosomes are defined as individual chromosomes that are closer to the spindle midzone than the mass of anaphase chromosomes. Chromosome bridges consist of a continuous mass of DNA that spans the spindle midzone with kinetochore signals oriented towards each pole (Review Fig. 1A). We are confident that this classification separates the two types of chromosome segregation errors but understand the advantages of having a quantifiable distinction.

While we did not measure spindle length, we offer a more detailed description of the chromosome bridges phenotype in Review Fig. 1B and C. Briefly, we marked the distance between anaphase chromosome masses (grey box) and draw a box that spans the central 33% of the spindle space (black box). We quantified DNA inside each box. We expected that lagging chromosomes will be identified by a low percentage of DNA inside the black box whilst anaphase bridges will show a high percentage of DNA inside the black box compared to total DNA signal. A similar type of analysis, albeit for uncongressed chromosomes, was performed in McGuinness *et al.*, *Curr Biol* 2009.

In Review Fig. 1C, we show that the ratio between inter-spindle space chromosome mass and the total chromosome mass is higher in *Wapl^{Δ/Δ}* oocytes than in *Wapl^{f/f}* oocytes. This shows that in *Wapl^{Δ/Δ}* oocytes it is more like to observe a chromosome mass in the inter-spindle space, which likely corresponds to chromosome bridges.

Fig. 1f is missing labeling on the last two panels (and Inserts is spelled Insects). We are sorry for these mistakes and have corrected them.

There is concern about the heavy reliance on indirect immunofluorescence for quantitative measurements of protein level, as stated by the authors (p. 7). I would suggest a quantitative western at a minimum to confirm the results from the staining.

We thank the reviewer for this suggestion and certainly appreciate that an independent orthogonal method greatly strengthens a result. In this case, we would like to point out the challenges of providing a quantitative Western blot of oocytes. First, we are only able to obtain small numbers of oocytes and therefore many mice have to be sacrificed to load at least 80 cells in a well. Secondly, depending on the abundance or rarity of protein of interest, possibly many more cells and mice are needed to detect it by Western blotting. Third, the biggest challenge for these particular experiments is that is not sufficient to load whole cell lysate. Instead, it is important to quantify chromatin-bound and not soluble cohesin. We are not aware that successful fractionation of such few cells has ever been reported. Hence, we hope the reviewer will appreciate our efforts to push the boundaries of our ability in achieving the desired Western blot.

We optimized the fractionation using 100 wild-type oocytes (Review Fig. 4A). We next attempted to perform the real experiment. We could isolate 80 oocytes from 5 females of *Wapl^{f/f}* and *Wapl^{f/f} (Tg)Zp3-Cre* each (litters from 3 breeding pairs each, since the likelihood of +/- Cre and female is 1:4) and fractionated these into soluble and chromatin-bound fractions. Due to the low material available, the signal to noise ratio is poor (Review Fig. 4B-

F). We initially stained the membrane with Rec8 and Scc1 primary antibodies, due to the low abundance of these cohesin subunits. Neither Rec8 nor Scc1 was detectable in the chromatin-bound fractions of either genotype, which we attribute to the low amount of α -kleisin subunits bound at this stage and the limited detection threshold (Review Fig. 4B and C). Smc3 was detectable in the soluble but not the chromatin-bound fraction of *Wapl^{f/f}* oocytes, consistent with a low abundance of cohesive cohesin and release of loop extruding cohesin by Wapl (Review Fig. 4D). Remarkably, chromatin-bound Smc3 was detectable in *Wapl^{Δ/Δ}* oocytes, even though just barely above detection limit. This is consistent with our findings from immunofluorescence stainings which revealed an accumulation of chromatin-bound cohesin upon Wapl depletion. However, the Smc3 signal is very weak for all conditions, indicating the poor quality of either the antibody or the membrane, which was at this point striped twice and stained three times. Unfortunately, we could not test the efficacy of the chromatin fractionation with H3 staining (Review Fig. 4F), likely do to the multiple restainings of the membrane. We could only show that GAPDH is present solely in the soluble fraction and not in the chromatin-bound fraction (Review Fig. 4E). Even though this experiment is far from perfect likely due to technical limitations, we hope the reviewer will appreciate our efforts to push the boundaries of our ability in trying to achieve the desired western blot.

The authors see an increase in chiasma-like structures (Fig. 2). Unfortunately, they make the conclusion that loss of Wapl leads to DNA breaks that are inefficiently repaired. This is highly speculative with very little evidence.

To provide evidence in support of this hypothesis, we performed *in situ* fixation of oocytes from *Wapl^{f/f}* and *Wapl^{f/f} (Tg)Zp3-Cre* females and examined breaks using the DNA damage marker phosphorylated histone H2AX (γ H2AX). Interestingly, we observed a significant increase of γ H2AX foci number in mature Wapl knockout oocytes, which indicates persistence of DNA damage. We have included these data in Fig. S2E and F. Therefore, Wapl loss leads to formation of DNA breaks. We do not know the mechanism for the production of the DNA breaks but one speculation is that torsional stress due to loop extrusion might facilitate these. If the DNA breaks are inefficiently repaired with a homologue-bias, then these structures might manifest as chiasma-like structures and lead to chromosome bridges in meiosis I and aneuploidy.

The authors then suggest that Wapl recognizes different pools of cohesin (existing, vs. newly synthesized). This was also very speculative, for which there is correlative data at best. They make a point that Wapl must release Scc1-cohesin based on bridges in Figure 3. (bridges spelled dridges). The authors must define and quantify these bridges.

I don't see how they can differentiate a bridge from a defect in compaction.

We appreciate the comments and would like to point out that our interpretation is based on epistasis experiments and not correlative data. We used conditional double knockout oocytes to test whether Wapl releases newly synthesized cohesin. In oocytes from *Wapl^{f/f} Smc3^{f/f} (Tg)Zp3-Cre* females, the conditional alleles are deleted during oocyte growth, which occurs long after cohesion established. Loss of Wapl leads to lack of cohesin release from chromosomes. If Wapl releases existing cohesin that was loaded during pre-meiotic S-phase,

then preventing new Smc3 synthesis in a Wapl Smc3 double knockout should have no effect and we should still observe an increase on cohesin levels. On the other hand, if Wapl releases newly synthesized cohesin, then preventing new Smc3 synthesis should prevent the increase in Smc3. Our results for *Wapl^{Δ/Δ}Smc3^{Δ/Δ}* and *Wapl^{Δ/Δ}Sccl^{Δ/Δ}* oocytes are both consistent with the latter, namely that new Smc3 and new Sccl synthesis is required for cohesin accumulation on chromosomes upon Wapl depletion (Fig. 3C-F).

We think that the Wapl phenotypes are due to the increase in chromosomal Sccl because the anaphase bridges are prevented in the Wapl Sccl double knockout oocytes (Fig. 3A and B). We have more accurately defined the bridges in the main text and performed a quantification that we include as Review Fig. 1.

The HiC data is used for additional speculations about cohesin position at the base of specific loops.

There appears to be a misunderstanding as we make no claims on cohesin position at the base of specific loops. We use Hi-C to quantify mainly loops and TADs and test whether they depend on Sccl cohesin and whether loop sizes increase upon Wapl depletion. We made some unexpected observations suggesting that there are differences between oocytes and somatic cells.

In summary, this manuscript addresses the role of Wapl in cohesin function in meiosis. The data lack quantification and analysis in the case of bridges and needs corroboration in the case of getting protein quantification from indirect immunofluorescence. The models in Figure 5 are only one of several that are consistent with the HiC data.

We have provided additional quantifications in the revised manuscript and described our efforts towards providing quantitative Western blotting of fractionated ultra-low input samples. We appreciate that our model may not be the only one that is consistent with Hi-C but it makes testable predictions. For example, it predicts that Rec8 is a barrier to loop extrusion and its removal should lead to longer loops, if cohesin release is prevented by Wapl knockout. We have tested this in Chatzidaki *et al.*, which we are including as supplementary material for the review process and found that Rec8 cleavage by TEV protease leads to longer loops. We therefore believe that the model presented in Fig. 5 fulfills the requirements of a model in making testable predictions, and it appears that some aspects of the model even appear to be correct.

Review figures Legends:

Review figure 1: Quantification of the chromosome segregation defects in *Wapl^{fl/fl}* and *Wapl^{ΔΔ}* oocytes and distinction between lagging chromosomes and chromosome bridges classification

(A) Schematic illustration of the different classes of chromosome segregation defects observed and quantified by live-cell imaging in figure 1C and D. Our classification is based on morphological criteria. Lagging chromosomes are defined as individual chromosomes that are closer to the spindle midzone than the mass of anaphase chromosomes. Chromosome bridges consist of a continuous mass of DNA that spans the spindle midzone with kinetochore signals oriented towards each pole.

(B) Schematic illustration of the quantification method used to quantify the chromosome segregation defects observed by live-cell imaging and distinguish between lagging chromosomes and chromosome bridges. The ratio between inter-spindle space chromosome mass (black box) and the total chromosome mass (grey box) was calculated for live-cell imaging movies of *Wapl^{ΔΔ}* and *Wapl^{fl/fl}* oocytes.

(C) Quantification of the ratio between inter-spindle space chromosome mass (black box) and the total chromosome mass (grey box) in *Wapl^{ΔΔ}* and *Wapl^{fl/fl}*.

Review figure 2: There is no progressive accumulation of Scc1 during the transition of immature to mature oocytes

(A) Smc1 levels were quantified in NSN (left) and SN (right) from *Wapl^{fl/fl}* and *Wapl^{ΔΔ}* oocytes. P-value = 0,0054 (SN) and = 0,0498 (NSN) using an unpaired t-test.

(B) Smc3 levels were quantified in NSN (left) and SN (right) from *Wapl^{fl/fl}* and *Wapl^{ΔΔ}* oocytes. P-value = 0,1067 (SN) and = 0,1847 (NSN) using an unpaired t-test.

Review figure 3: Rec8 is enriched in chiasma-like structures observed in *Wapl^{ΔΔ}* oocytes.

(A) Quantification of Rec8 levels in chiasma-like structures and natural chiasma observed on bivalents of *Wapl^{ΔΔ}* oocytes. The number of bivalents analyzed is indicated in the figure. P-value = 0,0039 using an unpaired t-test.

(B) Ratio between Rec8 levels observed on chiasma-like structures and on natural chiasma.

Review figure 4: Fractionation of GV oocytes and Western Blot analysis.

(A) Optimization of the chromatin-fractionation protocol using 100 wild-type oocytes. We confirmed that chromatin-fractionation worked properly by staining GAPDH (only present in the soluble fraction) and H3 (only present in the chromatin-bound fraction). WCL, whole-cell lysate; SF, soluble fraction; CB, chromatin bound fraction; W1 and W2, washing steps.

(B-F) Western Blot on fractionated *Wapl^{fl/fl}* and *Wapl^{ΔΔ}* oocytes stained with Scc1 (B), Rec8 (C), Smc3 (D), GAPDH (E) and H3 (F). Per sample 80 oocytes from 5 females per genotype were used. WCL, whole-cell lysate; SF, soluble fraction; CB, chromatin-bound fraction. Note that the H3 staining was performed last after multiple restainings of the membrane.

January 16, 2020

RE: JCB Manuscript #201906100R

Dr. Kikuë Tachibana
Institute of Molecular Biotechnology of the Austrian Academy of Sciences (IMBA)
Dr. Bohr Gasse 3
Vienna 1030
Austria

Dear Dr. Tachibana:

Thank you for submitting your revised manuscript entitled "Wapl releases Scc1-cohesin and regulates chromosome structure and segregation in mouse oocytes". We would be happy to publish your paper in JCB pending final revisions necessary to meet our formatting guidelines (see details below).

Please be sure to address the remaining minor comments voiced by reviewers #1 and #2 in your final revision (and please be sure to provide a point-by-point rebuttal to these reviews).

A. MANUSCRIPT ORGANIZATION AND FORMATTING:

Full guidelines are available on our Instructions for Authors page, <http://jcb.rupress.org/submission-guidelines#revised>. **Submission of a paper that does not conform to JCB guidelines will delay the acceptance of your manuscript.**

1) Text limits: Character count for Reports is < 20,000, not including spaces. Count includes title page, abstract, introduction, results, discussion, and acknowledgments. Count does not include materials and methods, figure legends, references, tables, or supplemental legends.

2) Figure formatting: Scale bars must be present on all microscopy images, including inset magnifications. Molecular weight or nucleic acid size markers must be included on all gel electrophoresis.

3) Statistical analysis: Error bars on graphic representations of numerical data must be clearly described in the figure legend. The number of independent data points (n) represented in a graph must be indicated in the legend. Statistical methods should be explained in full in the materials and methods. For figures presenting pooled data the statistical measure should be defined in the figure legends. Please also be sure to indicate the statistical tests used in each of your experiments (both in the figure legend itself and in a separate methods section) as well as the parameters of the test (for example, if you ran a t-test, please indicate if it was one- or two-sided, etc.). Also, since you used parametric tests in your study (e.g. t-tests, ANOVA, etc.), you should have first determined whether the data was normally distributed before selecting that test. In the stats section of the methods, please indicate how you tested for normality. If you did not test for normality, you must

state something to the effect that "Data distribution was assumed to be normal but this was not formally tested."

4) Materials and methods: Should be comprehensive and not simply reference a previous publication for details on how an experiment was performed. Please provide full descriptions (at least in brief) in the text for readers who may not have access to referenced manuscripts. The text should not refer to methods "...as previously described."

5) Please be sure to provide the sequences for all of your primers/oligos and RNAi constructs in the materials and methods. You must also indicate in the methods the source, species, and catalog numbers (where appropriate) for all of your antibodies.

6) Microscope image acquisition: The following information must be provided about the acquisition and processing of images:

a. Make and model of microscope

b. Type, magnification, and numerical aperture of the objective lenses

c. Temperature

d. imaging medium

e. Fluorochromes

f. Camera make and model

g. Acquisition software

h. Any software used for image processing subsequent to data acquisition. Please include details and types of operations involved (e.g., type of deconvolution, 3D reconstitutions, surface or volume rendering, gamma adjustments, etc.).

7) References: There is no limit to the number of references cited in a manuscript. References should be cited parenthetically in the text by author and year of publication. Abbreviate the names of journals according to PubMed.

Also, please note that we do not allow 'supplementary references' sections so please add any non-duplicate references from that list to the main references list.

8) Supplemental materials: There are strict limits on the allowable amount of supplemental data. Reports may have up to 3 supplemental figures. At the moment, you are below this limit but please bear it in mind when revising.

Please also note that tables, like figures, should be provided as individual, editable files. A summary of all supplemental material should appear at the end of the Materials and methods section.

9) eTOC summary: A ~40-50 word summary that describes the context and significance of the findings for a general readership should be included on the title page. The statement should be written in the present tense and refer to the work in the third person.

10) Conflict of interest statement: JCB requires inclusion of a statement in the acknowledgements regarding competing financial interests. If no competing financial interests exist, please include the following statement: "The authors declare no competing financial interests." If competing interests are declared, please follow your statement of these competing interests with the following statement: "The authors declare no further competing financial interests."

11) ORCID IDs: ORCID IDs are unique identifiers allowing researchers to create a record of their various scholarly contributions in a single place. At resubmission of your final files, please consider providing an ORCID ID for as many contributing authors as possible.

B. FINAL FILES:

-- High-resolution figure and video files: See our detailed guidelines for preparing your production-ready images, <http://jcb.rupress.org/fig-vid-guidelines>.

Thank you for this interesting contribution, we look forward to publishing your paper in Journal of Cell Biology.

Sincerely,

Arshad Desai, PhD
Monitoring Editor
Journal of Cell Biology

Tim Spencer, PhD
Executive Editor
Journal of Cell Biology

Reviewer #1 (Comments to the Authors (Required)):

This was already an interesting paper, and the revisions have sufficiently addressed my concerns.

Publication is recommended.

One minor point that can be addressed with some extra explanation:

Regarding the gH2AX staining in the mature oocytes, it was not entirely clear to me whether these cells had undergone any division without WAPL. The explanation given to the referees is that this gH2AX is due to torsional stress. But couldn't this also be due to a prior segregation error?

Explaining or rephrasing would be fine here.

Reviewer #2 (Comments to the Authors (Required)):

Overall the authors have addressed my concerns and in my opinion this study should be published in J Cell. Biol.

The text on page 7 relating to Figure 3C-F is still quite incomprehensible. I understand the point the authors want to make, but I think this has to be explained better- maybe the authors can add part of the text from the rebuttal letter, and a scheme explaining the rationale of the experiment?

On the side, I think the effort the authors made to detect the different fractions of cohesin subunits in *Wapl* f/f and knock-out oocytes by western blot is very impressive, given the low amounts of protein per cell, and the small amount of oocytes per mouse. I do not think that quantification by western blot of cohesin subunits in oocytes can or should be a condition for acceptance of this manuscript.

Reviewer #3 (Comments to the Authors (Required)):

The reviewers have adequately addressed the reviewers comments.